# LLMs as a Knowledge Source for Procedural Video Understanding

## Abstract

This paper addresses the challenge of understanding and interpreting human activities in procedural video content. Unlike previous approaches that rely solely on human annotations, we propose to leverage the vast amount of information stored in large language models (LLMs) to improve the prediction capabilities of vision models for procedural videos. Our framework uses LLMs to extract detailed procedural instructions and contextually relevant information to enhance the training process of video models. We demonstrate that this methodology not only refines the model's ability to predict complex hierarchical human activities but also extends its zero-shot capabilities, allowing it to generalize to unseen activities, as well as across hierarchical levels. Our simple approach outperforms baselines across different activity recognition tasks and datasets, demonstrating the benefits of exploiting the structured knowledge within LLMs.

## 1 Introduction

Understanding human activities from video is an important research area with substantial applications in fields such as robotics or human computer interaction. Procedural activities, which involve multiple connected steps towards achieving a goal, are particularly challenging as they require a deep understanding of temporal and hierarchical human activities (Bandura, 2001; Cooper & Shallice, 2006). The process of baking bread, for example, requires understanding that the dough has to be prepared before the bread can be put in the oven. It also requires understanding that preparing the dough is a step that in turn requires multiple substeps, such as mixing the ingredients, kneading the dough, and letting it rise. On top of this, there are other contextual dependencies for each of these steps, such as an environment (*e.g.*, kitchen), tools (*e.g.*, oven), and reason (*e.g.*, if the dough does not rise, the bread will not cook properly). Procedural video understanding requires all of this information to be present in the system to properly assess the activity taking place in the video.

While in recent years researchers have attempted to extract this procedural knowledge from raw video data (Lan et al., 2015; Wang et al., 2018; 2016; Zhao et al., 2017; Tang et al., 2012), the process has proven challenging and largely unsuccessful. An alternative method has been to use either Internet-sourced text, such as narrations from online videos (Miech et al., 2019; 2020) and relevant text from web knowledge bases (*e.g.*, wikiHow) (Lin et al., 2022; Chen et al., 2022; Mavroudi et al., 2023), or to include manually added human annotations (Grauman et al., 2022; Song et al., 2023). However, each method presents its own set of challenges. Internet-sourced text is often far too noisy (Miech et al., 2020; Han et al., 2022b; Mavroudi et al., 2023), whereas manual annotations, although more precise, can be costly to collect in substantial quantities, especially those that effectively depict the temporal and hierarchical interconnections between actions.

In this paper, we propose to source this procedural knowledge not from the videos themselves or from the Internet, but from large language models (LLMs), harnessing their capabilities as knowledge synthesizers. LLMs are trained to complete text sentences, primarily obtained from the Internet (Touvron et al., 2023; Devlin et al., 2018; Brown et al., 2020). This allows such models to adopt and store a plethora of information and knowledge found online, enabling the prediction of similar content in the future.

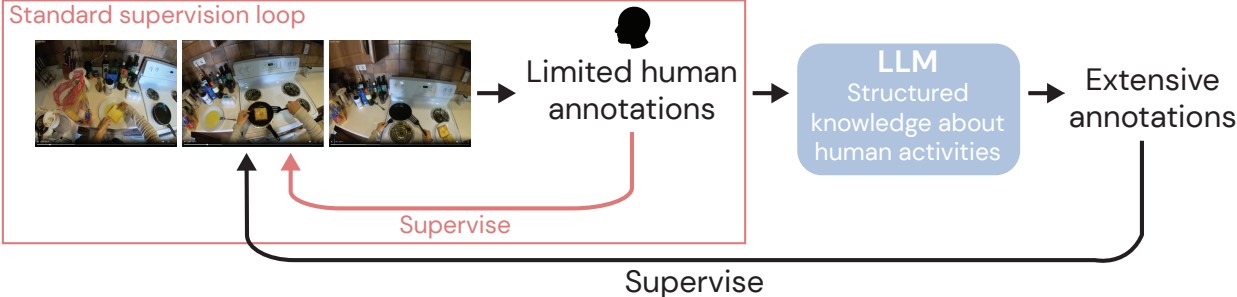

Figure 1: We exploit the procedural activity knowledge stored in LLMs in order to improve the training process of video models.

One major category within this knowledge is the comprehensive procedural instructions for various human activities found in online articles and blogs. Considering this, we propose to harness the existing knowledge within LLMs about procedural activities (Huang et al., 2022; Yuan et al., 2023; Gramopadhye & Szafir, 2022) in order to train a vision model. The focus is on exploring the untapped potential of such models in understanding procedural videos—a process that requires more intricate annotations or knowledge compared to atomic action recognition. It is worth emphasizing that obtaining such a complete set of annotations via human annotation is exceptionally difficult.

This paper makes the following contributions. First, we propose a framework that makes use of existing annotations in procedural video datasets to prompt large language models to deliver more extensive information about the video, both in breadth and in depth, and incorporates this LLM-obtained information to the training of a video model. Second, we demonstrate that our framework improves prediction metrics in complex hierarchical human activities across different activity recognition tasks and datasets, for both egocentric (Song et al., 2023) and exocentric (Zhou et al., 2018) videos. Finally, we provide an extensive analysis on the zero-shot capabilities of such a model, that are inherited from the open-domain knowledge of the LLM. Our framework improves performance across unseen activity categories, as well as unseen activity hierarchical levels.

## 2 Related Work

**Procedural Activity Recognition** The study of procedural human activities has been the focus of several works in recent years. Procedural activities require a deeper understanding of goals, motivations and interrelations between actions compared to what is needed for atomic action recognition. To facilitate this research, various datasets (Zhou et al., 2018; Grauman et al., 2022; Song et al., 2023; Sener et al., 2022; Tang et al., 2019; Zhukov et al., 2019; Shao et al., 2020; Zala et al., 2023; Ben-Shabat et al., 2021; Maeda et al., 2024) have been collected. Ego4D (Grauman et al., 2022), in particular, has emerged as the most comprehensive and challenging one. By focusing on first-person egocentric videos, it accentuates the significance of long-term human activities, thus emphasizing the procedural nature of the dataset. Building on top of Ego4D, Goal-Step (Song et al., 2023) further annotates videos with dense goal, step, and substep labels.

The tasks posed in these datasets have been approached in different ways. When the temporal boundaries of steps are not provided, soft constraints on the order of actions are enforced (Bojanowski et al., 2014; 2015; Zhu et al., 2015; Tapaswi et al., 2014; 2015; Alayrac et al., 2016; Lin et al., 2020; Papadopoulos et al., 2022; Salvador et al., 2021; Lin et al., 2022; Dvornik et al., 2022; Chen et al., 2022; Mavroudi et al., 2023; Lin et al., 2022), based on non-parametric knowledge bases, such as collections of wikiHow articles, that are general to the activity and not instance-specific. These works, however, require finding for each video a corresponding instructional article. This video-article matching is error-prone and thus may lead to inaccurate supervision. Even when the correct instructional article is found, its content may not fully match the video. The advantage of using LLMs is that they can dynamically synthesize knowledge that is specific to each video instance on the basis of the available annotations.

Grammar parsers, either fixed or learned, have also been applied to enforce structured predictions (Xu et al., 2022; Qi et al., 2018; 2020; Ling & Geng, 2019; Su et al., 2017; Vo & Bobick, 2016; Jia, 2022; Luo et al., 2022b). However, most recent procedural datasets contain temporal annotations, either noisily obtained from the Internet, and thus weakly correlated (*e.g.*, HowTo100M (Miech et al., 2019)) or obtained via clean human annotation (*e.g.*, Ego4D (Grauman et al., 2022) or Goal-Step (Song et al., 2023)). In such cases there is a direct mapping from the video segments to their corresponding annotations (or sets of annotations). This allows approaches to train either using contrastive learning techniques (Han et al., 2022a; Dvornik et al., 2023; Ashutosh et al., 2023; Zhao et al., 2022; Ko et al., 2022), or directly in a supervised manner (Song et al., 2023; Zhang et al., 2022; Xu et al., 2021b; Shen & Elhamifar, 2024; Kim et al., 2024). We combine these two approaches.

**Video-Language Pretraining** The learning of models that combine textual and video information has been approached in the literature in two different settings. In the representation learning setting, visual and text embeddings are projected into a shared feature space using metric learning techniques (Weston et al., 2010; Frome et al., 2013). These representations are later used for retrieval tasks (Xu et al., 2021a; Luo et al., 2022a; Fu et al., 2021; Bain et al., 2021; Miech et al., 2020; Ge et al., 2022; Wang et al., 2022a; Xue et al., 2022; Lei et al., 2021; Zhao et al., 2022).

In the generative setting, video and text inputs are combined in an early fusion fashion, and the model is trained using next token prediction, typically using decoder-only Transformer architectures (Vaswani et al., 2017). Such architectures have been used for several tasks including captioning, action recognition, or video question answering (Sun et al., 2019b; Su et al., 2019; Sun et al., 2019a; Yu et al., 2023; Weng et al., 2024). These two settings are not mutually exclusive (Yan et al., 2022; Kumara Kahatapitiya et al., 2023; Yang et al., 2022). In our paper, we adopt the representation learning setting, which is closest to the approaches that train on categorical labels.

**LLM's Knowledge for Procedural Video** The literature has leveraged the procedural knowledge of large language models (LLMs) in various ways. Some studies (Zeng et al., 2023; Wang et al., 2022b) have employed LLMs for video prediction tasks focusing on reasoning solely within the textual domain. Additionally, LLMs have been utilized as an initial step to determine the processing steps required for video analysis (Surís et al., 2023). Earlier research has also involved curating domain-specific text, training language models with this data, and subsequently applying the acquired knowledge to videos (Sener & Yao, 2019). LaViLa (Zhao et al., 2022) uses LLMs to augment video annotations, however it requires training a video-text captioning model, and it only annotates videos with descriptive captions, not providing any other information about the activity taking place. More related to our approach is AntGPT (Zhao et al., 2023), which predicts human goals through in-context learning using LLMs, but it focuses on just predicting action labels. Instead, our approach leverages the extensive open-domain knowledge that LLMs possess about human activities.

## 3 Method

To train our video model accurately, we need to capture the full scope of each human activity: not just their labels, but also the visuals, the context, the sequential and hierarchical nature of the activity, the reason behind each one of the steps, and so on. Take making bread as an example: it involves knowing what each ingredient does, how they are mixed, or why the oven has to provide consistent heat. Each piece of this knowledge not only provides visual or contextual cues for identifying an activity or its individual steps, but also enhances our ability to recognize the activity by enabling us to link related concepts. For example, understanding the role of an oven in bread-making aids in associating not only the oven but also tools that offer similar functionality, such as a heavy pot, with the process of baking bread.

While humans have this information as commonsense knowledge, annotating every aspect of large video collections with such detail can be prohibitively expensive. Fortunately, large language models already possess this information, and they can provide it from labels like "bake bread," if queried appropriately. Our goal is to distill this comprehensive knowledge from LLMs into our video model, using the existing video annotations as the bridge between them.

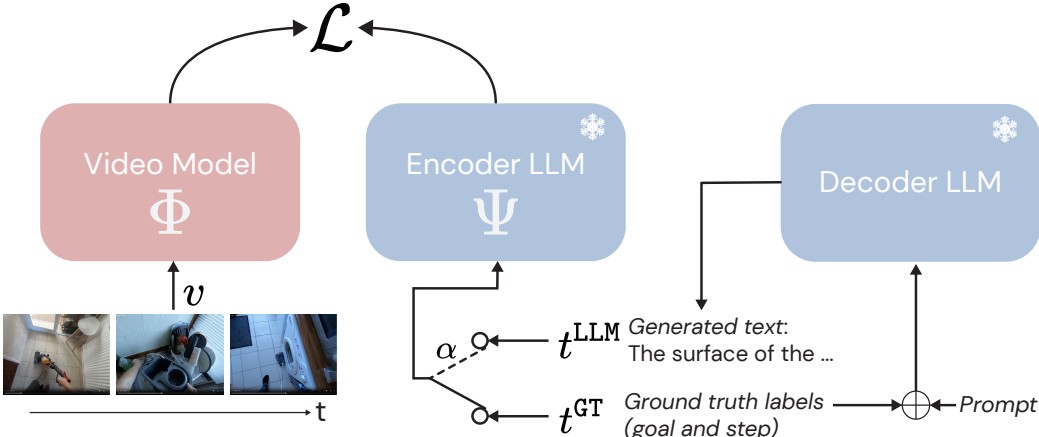

Figure 2: **Schematic of the architecture.** We use pink to represent learnable parameters, and blue to represent frozen parameters.

Specifically, we first extract this information from the LLM, and then use it to supervise a video-text model. The prompting to the LLM, discussed in section 3.1, is a crucial part of our approach. After training, the model can be used for video recognition tasks, not requiring any extra knowledge provided at inference time. The overall procedure is illustrated in fig. 2.

## 3.1 Extracting the Knowledge From LLMs

Based on prior work, we define five knowledge categories that language models can provide information about. For each category, we manually design text prompts to query the language model about them, given the available annotations. The prompts assume goal (activity label, *e.g.*, make omelet), step (sub-action within the activity, *e.g.*, prepare egg mixture) and substep (sub-action within the step, *e.g.*, crack eggs into a bowl) annotations. If substep annotations are not present, the prompts associated with them are not used. These five knowledge categories are:

**1) Pre- and post-condition**. Certain actions can only be performed when specific conditions are met. For example, in order to sauté food, it needs to be cut into small, even pieces. Similarly, some actions are only finished when they meet certain conditions. For example, the action of roasting vegetables is only considered complete when the vegetables are tender and display the expected amount of browning. Prior work (Wu et al., 2023) has shown that procedural video understanding can benefit from such information. In our scenario, these conditions provide additional contextual cues to recognize actions, by looking at previous and later steps. We define four prompts under this category: two for each step's conditions (pre- and post-), and likewise, two for substeps.

**2) Partonomic action hierarchy**. Actions are *part* of longer "parent" actions and, at the same time, can be decomposed into shorter "children" sub-actions or steps. For example, grilling vegetables is part of cooking dinner, and it can be decomposed into other sub-actions like heating the grill. Prior work highlights the benefits of embedding hierarchical structure within predictive models(Ashutosh et al., 2023; Mounir et al., 2023; Wu et al., 2022; Gong et al., 2023; Surís et al., 2021). We define five different prompts under this category: three that query for possible children of the provided goals, steps, and substeps of the activities, and two more that query for parents of the provided steps and substeps.

**3) Visual scene**. Actions take place in a scene where certain visual elements are present. Such visual elements can be, for example, the objects required to perform the action, or physical locations such as a kitchen for cooking or a swimming pool for swimming. The static visual context has been proven very useful—sometimes even sufficient—to solve video recognition tasks (Buch et al., 2022; Lei et al., 2022; Yu

**Rephrase goal label**

"In the context of human activities, there is one task that has as a goal to {goal_label}. What is an equivalent way (or synonym) of expressing this goal? (Be brief, provide just one specific answer, do not explain the reasoning behind it, and remain silent after answering) An equivalent way of expressing {goal_label} is "

**Post-conditions of substep**

"In the context of human activities, we are performing a task with the goal of {goal_label}, and we are currently in the step of {step_label}. The next substeps are {posterior_substeps}. We have just done the substep of {prior_substeps[-1]}. Think about the prior substep and about the conditions that are required now to guarantee that the previous substep has been finished. What are the visual attributes of the current scene that make us know that the prior substep {prior_substeps[-1]} has finished successfully? The post-conditions after finishing the substep {prior_substeps[-1]} are (define them by describing the state of the current scene; remain silent after answering): "

**Parent goal**

"In the context of human activities, we are performing a task with an unknown goal. We have done the steps of {prior_steps}, and are now in the step of {step_label}. The next steps are {posterior_steps}. Given these steps, what is the overall goal of the task? (Be brief and remain silent after answering) The goal is to "after answering): "

Figure 3: **Prompts for the LLM**. Some examples of prompts that we input to the LLM. We show three out of a total of 17, as described in section 3.1, and share the rest in appendix A. Given a query template like the ones in this figure, the query is created by replacing the placeholders (e.g. "goal label") with the actual labels (e.g. "bake").

et al., 2023). We define three different prompts under this category, corresponding to each hierarchical level of the provided ground truth labels.

**4) Reason for the action**. In the context of a goal-oriented activity, every step has a deliberate reason behind it (Bandura, 2001). For instance, the purpose of marinating meat is to enhance its flavor, tenderness, and moisture. When the model is trained to understand the purpose behind an action in a given step, it can later connect that action with others that share the same objective. In the example above, the model will learn to connect the action of "marinating" with the action of "seasoning with herbs and spices," as they share the purpose of enhancing the flavor of the food. Understanding the reason behind a step can additionally link that step with specific visual information. For example, if the reason to add yeast is for the dough to rise, the model will associate "add yeast" with "risen dough." Intent- or goal-oriented actions have been studied in a learning context both from the computational (Lieberman & Willcox, 2012; Reuss et al., 2023; Nasiriany et al., 2019) as well as the neuroscience perspectives (Blakemore & Decety, 2001; Goldman, 2015). Ablations in section 4.5 show that it is the most important piece of additional knowledge.

**5) Synonyms**. Actions can be referred to in different ways, either using synonyms, or using longer phrases. This can be used as an additional signal during training. Paraphrasing annotations has been used in the procedural video literature with great success (Zhao et al., 2022), as it helps connect different language formulations to the same visual content, as a sort of data augmentation. For each goal, step, and substep instances, we query the LLM for a synonym, given the context of the rest of annotations.

We provide examples of prompts in fig. 3, and share all prompts in appendix A. Note that the contribution of this paper is orthogonal to any prompt engineering techniques (Gu et al., 2023; Yang et al., 2023b). For the purposes of this paper, we used fixed prompts, and did not optimize them. Prompts were curated independently of the domain: we froze them before running any experiments, so they do not overfit to the datasets. Potentially, even better results could be obtained by refining the prompts following prior work, for example by using in-context examples.

### 3.2 Language Model

We use LLMs in two different modes: as an encoder and as a decoder. In the decoder mode, both the input and the output of the LLM are text. This mode is used to extract the knowledge present in LLMs: we input the available human annotations (*e.g.*, goal and step labels), together with a pre-determined prompt with instructions, and the LLM outputs additional information about the scene.

**Provided:**
**Goal category:** "Make French toast"
**Step category:** "Pan fry eggy bread slice to golden brown"

**Information added by LLM:**
**Rephrase of step:** "Cook eggy bread in pan"
**Negative:** "Add oil to the pan"
**Visual description:** "The stovetop is clean and clear of any debris, with a pan or pot sitting on top, containing a sizzling golden brown eggy bread slice, and a spatula resting on the side of the pan; the kitchen is warmly lit with overhead lighting, and the aroma of toast and eggs fills the air; the kitchen countertops are organized and clear of clutter, with a neat stack of clean plates"
**Pre-condition:** "The pan or pot is preheated and ready to use. The eggy bread slices are ready to be fried. The cooking oil or butter is melted and ready to use"
**Post-condition:** "1. The bread slices are now golden brown and crispy on both sides. 2. The eggs are cooked through and the egg white is set. 3. The pan is empty except for the cooked bread slices and eggs. 4. The stove is turned off. 5. The kitchen is filled with the aroma of cooked"

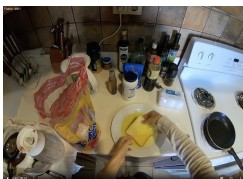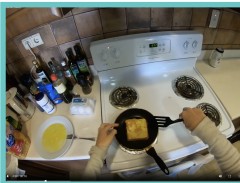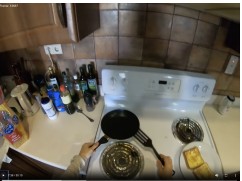

**Provided:**
**Goal category:** "Make taco"
**Step category:** "Make sauce"
**Substep category:** "Stir and mix ingredients on kitchenware to cook: Stir sauce"

**Information added by LLM:**
**Parent step** *(step and goal categories are not provided, just substep)*: "Prepare sauce"
**Reason for step:** "We want to mix the ingredients together so that they are well combined and heated through. This will help to ensure that the salsa is flavorful and has a good texture. Additionally, stirring the sauce will help to prevent it from burning or sticking to the bottom of the pan."
**Visual description:** "The scene is of a person standing in front of a stove, with a pot of salsa on the burner, a wooden spoon in their hand, and a look of concentration on their face as they stir the salsa, tasting it occasionally and adjusting the seasoning as needed."
**Post-condition of substep:** "1. The sauce is now well mixed. 2. The sauce is now at a consistent temperature throughout. 3. The sauce has a uniform texture. 4. The sauce is now at the desired consistency. 5. The stirring utensil is now clean and free of sauce."

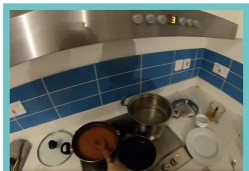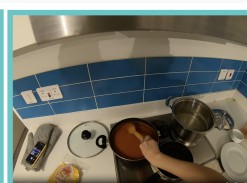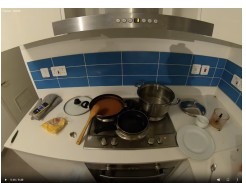

**Provided:**
**Goal category:** "Make tiramisu"
**Step category:** "Make custard mixture"

**Information added by LLM:**
**Reason for step:** "We are measuring the ingredients to ensure that we have the correct proportions of each ingredient, which is crucial for the taste, texture, and overall quality of the tiramisu."
**Substeps:**
"To measure ingredients, the following substeps are required:
1. gather all ingredients
2. read recipe to determine measurement requirements
3. use measuring cups or spoons to measure ingredients
4. record measurements for reference
5. ensure measured ingredients are within expiration date
6. check for any discrepancies"

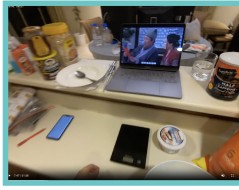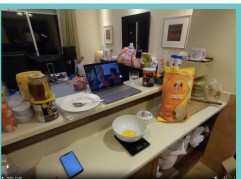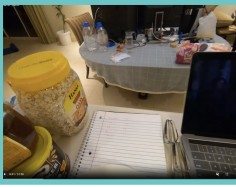

**Provided:**
**Goal category:** "Clean kitchen appliances"
**Step category:** "Wash utensils in kitchen sink"

**Information added by LLM:**
**Parent goal** *(goal category is not provided)*: "Organize the kitchen"
**Post-condition of step:** "You have finished washing the dishes and have dried them. The kitchen sink is empty. The dishes are put away in their proper places. The kitchen countertops are clear of any clutter. The stove, oven, and refrigerator are clean. The floor is swept and mopped. The trash cans are empty. The kitchen is tidy and organized."
**Substeps:** "A. Turn on the faucet. B. Add dish soap. C. Scrub the dishes. D. Rinse the dishes. E. Dry the dishes. F. Put away the dishes."

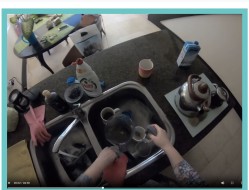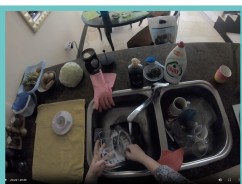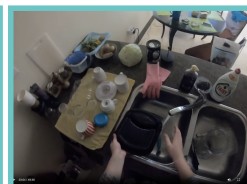

Figure 4: **Examples of predicted text**. The frames in teal correspond to the specific step or substep the provided annotations correspond to. The other frames are used to show how well the pre- and post-conditions match the video. We show some prompts in fig. 3.

We then encode by projecting the text to a learned embedding space. This setting—separate encoding and decoding— allows us to treat the text predictions of the LLM the same way we treat the already existing human labels: encoding them to a visual-language space. We represent the ground truth annotations with $t^{\texttt{GT}}$, and the LLM-generated annotations with $t^{\texttt{LLM}}$. The language encoder $\Psi$ computes embeddings $\Psi(t) \in \mathbb{R}^D$ for both of these inputs. In both modes, we freeze the weights of the LLM, which allows us to pre-compute both the decoder predictions as well as the text embeddings before training.

### 3.3 Video Model

The video model $\Phi$ takes as input precomputed features $v$ of a video clip and outputs a vector $\Phi(v) \in \mathbb{R}^{L \times D}$, where $L$ is the length of the video and $D$ is the embedding dimension. The output dimension $D$ is the same as the output of the language model. We train the video model from scratch, and assume that the extracted features preserve a sufficient level of temporal granularity to detect changes in steps within an activity.

### 3.4 Using the Knowledge

To supervise the video model with LLM-generated annotations, we project both the video and textual representations to a joint space and compute a similarity score $s$. This makes the adaptation to methods that already rely on shared feature spaces (Xu et al., 2021a) seamless, the only modification during training being the incorporation of LLM-generated annotations. For methods that rely on the vanilla (supervised) cross-entropy, we propose a minimal alteration: we replace the last linear classification layer with a dot product operation. This operation is computed between the visual features, which would typically be fed into the linear layer, and the textual embeddings of the labels. We then apply the cross-entropy loss in the standard fashion, which in our context translated to the InfoNCE loss (Oord et al., 2018; Khosla et al., 2020). This allows us to replace the labels with any other textual annotation. Specifically, we minimize the following loss function:

$$\mathcal{L} = \sum_{i=1}^{N \times L} \frac{s_{ii}}{\sum_j s_{ij}}, \quad \text{where} \quad \begin{aligned} s_{ij} &= (1-\alpha) \exp\left(\Phi\left(v_i\right)^{\intercal} \Psi\left(t_j^{\texttt{GT}}\right)\right) + \\ &\quad \alpha \exp\left(\Phi\left(v_i\right)^{\intercal} \Psi\left(t_j^{\texttt{LLM}}\right)\right) \end{aligned}, \tag{1}$$

where $\alpha$ controls the ratio of LLM-generated texts with respect to ground truth texts and $N$ is the number of videos in the dataset. In practice, we optimize eq. (1) in a mini-batch setting, and instead of incorporating $\alpha$ in the loss, we control it by sampling LLM-generated texts $(100 \times \alpha)\%$ of the times. We set $\alpha = 0.5$, see fig. 8 for ablations. The language annotations correspond to specific temporal spans of the video, and they are considered positives for all samples within that span. For samples containing more than one LLM-generated annotation, we randomly sample one of them to obtain $t^{\texttt{LLM}}$.

## 4 Experiments and Analysis

### 4.1 Datasets and Tasks

Our approach is specifically designed for tasks that require a holistic understanding of procedural videos. Therefore, its effectiveness increases in parallel with the degree to which procedural video understanding is required. Consequently, we focus our evaluation on benchmarks for hierarchical action recognition, as these benchmarks inherently demand a comprehensive understanding of the activities and can significantly benefit from additional contextual information. We show the generality of our method by evaluating on two distinct datasets and settings.

**Goal-Step** Goal-Step (Song et al., 2023) is a recent dataset that labels a portion of Ego4D (Grauman et al., 2022) videos with goals, steps, and substeps, thus being perfectly suited for our experiments, as it is procedural and contains a comprehensive hierarchy of actions. In the context of procedural videos, we argue that the online setting—where the system must make predictions based solely on past data—is particularly interesting: the information available is by design only partial (no future observations), emphasizing the need

for a complete understanding of activities and the relationships between actions. Therefore, we use Goal-Step to evaluate online activity recognition. Following the original paper, we evaluate activity recognition using mean average precision (mAP).

**YouCook2** We also evaluate in the offline setting on YouCook2 (Zhou et al., 2018), which contains steps for every activity. We adopt the same training and evaluation setting as the VideoCLIP (Xu et al., 2021a) paper, and evaluate text-video retrieval using Recall@$K$, which measures the proportion of positive activities that are successfully retrieved within the top $K$ results, as well as Median Rank. YouCook2 contains fewer hierarchical annotations than Goal-Step, so fewer options for LLM prompts are available. Specifically, the prompts requiring sub-step information are not used.

## 4.2 Implementation Details

For simplicity, we use the same language model for both the encoder and the decoder LLMs. Specifically, we use the open-sourced Llama2-70B-chat (Touvron et al., 2023).

**Goal-Step** For online activity recognition we use LSTR (Xu et al., 2021b), which has been proven effective at modeling the long-term context of online sequences, as well as being the model used in Song et al. (2023). Following the results reported in Song et al. (2023), we report goal prediction results using Omnivore (Girdhar et al., 2022) features and step prediction results using EgoOnly (Wang et al., 2023) features.

**YouCook2** For offline experiments, we also build on prior work and use the VideoCLIP (Xu et al., 2021a) architecture, as well as the S3D (Miech et al., 2019) features they use. In this paper, we specifically address the problem of learning from only the limited annotations available in the dataset, so unlike prior work (Xu et al., 2021a), we train the model from scratch.

## 4.3 Results

**Baselines** Weak label supervision has been used in video-text models in various settings. We choose captioning (Zhao et al., 2022) and paraphrasing (Yoo et al., 2021) as our baselines for their popularity in the literature (Yang et al., 2023a; Zhao et al., 2022; Seo et al., 2022; Raffel et al., 2020; Li et al., 2022). The captioning approach generates descriptions of a video using a captioning model, and then uses those generated descriptions as pseudo-labels to supervise a video-text model. We use LaViLa (Zhao et al., 2022) to implement this baseline. The paraphrasing approach consists in rephrasing the existing labels using different NLP tools (Yoo et al., 2021), which can be seen as a form of data augmentation, and it is equivalent to the "synonyms" subset of our method, for which we report the numbers. The key difference with respect to our approach is that in a paraphrasing scenario no new concepts or knowledge are provided to the model. We exploit the fact that LLMs store knowledge, and incorporate it into the training.

Additionally, we report the results when training solely with the ground truth labels ($\alpha = 0$) in a completely supervised setting, which we call "baseline." In both evaluation scenarios, our baseline is based on prior work, with the only distinction from our method being the data used for training: in addition to the ground truth annotations, we enhance our dataset with LLM-generated data, obtained from those annotations. All other training details remain the same, ensuring that the efficacy of our approach is cleanly isolated and measurable. Regardless of the value of $\alpha$ and the origin of $t^{\text{LLM}}$, all experiments are based on the same number of gradient updates; only the text signal $t$ changes, making the experiments fair and comparable.

**Goal-Step** We show results on Goal-Step in table 1. The first two rows correspond to the setting where only ground truth annotations are used. We only report the best-performing baselines in Song et al. (2023), and not the lower-performing architectures. The InfoNCE version is a reformulation of the standard cross-entropy loss, which adapts the method in Goal-Step (Song et al., 2023) to our formulation, in order to have an apples-to-apples comparison with our method. Note that this reformulation performs on par with its cross-entropy counterpart, which confirms the validity of the reformulation. The results show a substantial improvement when complementing the training with LLM-generated annotations, both for goal and step prediction—36% and 13% relative increase in mAP, respectively. These results set a new state-of-the-art on this benchmark. Our approach also outperforms the captioning and paraphrasing approaches in both the

Table 1: **Goal-Step online detection results**. We report the results on mAP for goal and step (which includes step and substep instances) online predictions on the Goal-Step dataset (Song et al., 2023). [†]The test set of goals is very small and thus very noisy. We include it here for completeness.

| | **Goal** (Omnivore features) | | **Step** (EgoOnly features) | |
| --- | --- | --- | --- | --- |
| | Seen | Zero-shot[†] | Seen | Zero-shot |
| GoalStep (Song et al., 2023) w/ CE | 24.5 | - | 10.8 | - |
| GoalStep (Song et al., 2023) w/ InfoNCE (Baseline) | 25.9 | 6.4 | 10.5 | 3.0 |
| Captioning (Zhao et al., 2022) | 30.1 | **21.9** | 11.1 | 3.0 |
| Paraphrasing (Yoo et al., 2021) | 35.1 | 9.1 | 11.3 | 3.9 |
| Ours | **35.2** | 11.0 | **11.9** | **7.2** |

Table 2: **YouCook2 results**. We train a VideoCLIP (Xu et al., 2021a) model from scratch, and report Recall@$K$ numbers (as a percentage) and Median Rank (MR). The baseline corresponds to VideoCLIP trained from scratch on the ground truth YouCook2 data.

| | **R@1** ↑ | **R@5** ↑ | **R@10** ↑ | **MR** ↓ |
| --- | --- | --- | --- | --- |
| Baseline | 11.3 | 29.8 | 41.9 | 16 |
| Captioning (Zhao et al., 2022) | 13.2 | 34.3 | 46.9 | 12 |
| Paraphrasing (Yoo et al., 2021) | 13.0 | 33.3 | 45.6 | 13 |
| Ours | **14.6** | **37.1** | **50.7** | **11** |

"seen" and "zero-shot" categories, as well as being better in almost all other splits. We discuss ablations in section 4.5 and zero-shot results in section 4.4.

**YouCook2** The results in table 2 show a significant contribution of the LLM-generated prompts. Our approach outperforms the baseline ($\alpha = 0$) by a significant margin, as well as being significantly better than the captioning and paraphrasing approaches. Note that better results (32.2 R@1, 62.6 R@5 and 75.0 R@10) can be obtained when pretraining on large-scale auxiliary datasets (Xu et al., 2021a).

### 4.4 Few-Shot and Zero-Shot Capabilities

An important advantage of leveraging the language space of strong LLMs as the target space for video prediction is their ability to represent language beyond the specifics of the training dataset. Provided that the video model has learned a good mapping from the input frames to this language space, it will be able to recognize infrequent (few-shot) of even novel (zero-shot) categories at inference time. We study these capabilities in two different settings.

**Generalizing to Novel Actions** The testing set of Goal-Step contains goals and steps not present in the training set, as well as classes with very few samples. Table 1 shows that in the zero-shot scenario, the incorporation of LLM-generated annotations improves performance with respect to the initial baseline from Song et al. (2023), both for goal and for step predictions. While some baselines perform better at the goal level, we attribute this to the small size of the goal-level zero-shot test set, which adds some noise to the evaluation. The step-level zero-shot test set contains a larger number of examples, as well as being a more relevant metric for fine-grained activity recognition.

We further show this behavior in fig. 5, where we plot the relative mAP improvement per class, as a function of the number of training occurrences for that class. We combine classes in groups of 100 occurrences, and plot the average for every group. The analysis shows that our approach is more beneficial the rarer the classes are during training. The exponential fit to the ungrouped data provides the same conclusions.

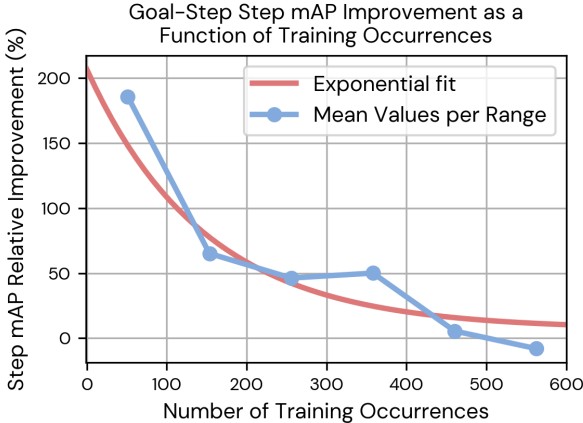

Figure 5: **Generalization to rare classes**. We plot the per-class relative improvement of our method with respect to the baseline, as a function of the number of training occurrences. Each point represents the average over all the classes within an occurrence range, in groups of 100 occurrences. The rarer the class is (fewer number of training occurrences, shown to the left of the x-axis), the more useful our approach becomes. See section 4.4.

Table 3: **Generalization across hierarchical levels**. We study our models' transferability across Goal-Step hierarchical levels. The arrows represent "Trained on → Tested on".

|  | **mAP (%) ↑** | |
|  | Goal → Step | Step → Goal |
|---|---|---|
| Baseline | 0.5 | 8.1 |
| Ours | **2.4** | **13.6** |

**Learning from Other Hierarchical Levels**  In table 3 we analyze the capabilities of our model when trained using Goal-Step labels that do *not* correspond to the hierarchical activity level we train with. This is, how much step-level ("subaction") labels can inform goal-level ("action") predictions, and vice-versa. Our approach significantly improves mAP results in both cases.

### 4.5  Contribution of Knowledge Categories

We analyze the contribution of each knowledge category used in our experiments. Figure 6 reports ablations where, rather than training with all the LLM-generated annotations, we train with the annotations obtained from specific knowledge categories. The results suggest that including visual knowledge is more useful in YouCook2, where the procedural component is not as strong. In contrast, it does not contribute as much in Goal-Step, where higher-level, more open-ended and procedure-related knowledge categories, such as "condition" or "hierarchy" take a more relevant role. All the rest of knowledge categories have a positive contribution compared to the baseline.

Upon examining fig. 6, one might wonder if *any* form of additional information, as long as it is based on the ground truth labels, would contribute to the performance of our approach. To explore this idea, we created a different type of annotation, where we instructed the LLM to compose a poem about the ground truth activity. Although the poem reflects the activity, the nature of the prompt does not guide the LLM to provide targeted insights that might be applicable to procedural tasks. The results in fig. 6 indicate that the mere presence of more text annotations is not the key to the method's success. The additional data must provide relevant insights pertaining to the activity in question.

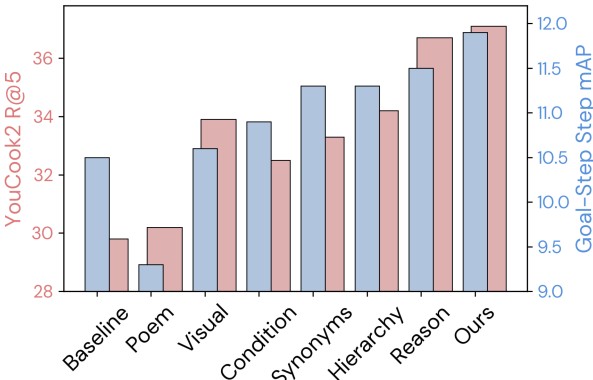

Figure 6: **Ablations**. We report experiments where the only LLM-generated annotations come from each one of the knowledge groups defined in section 3.1. See section 4.5 for analysis.

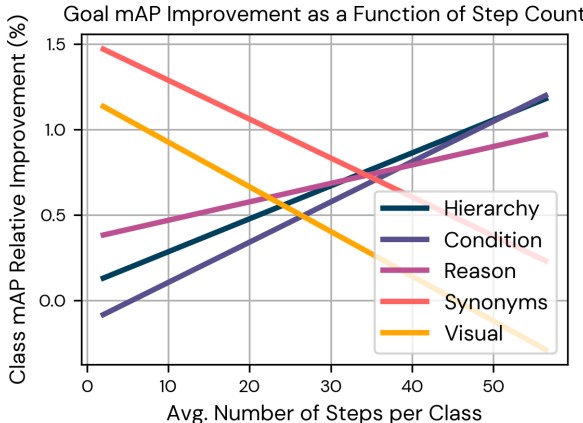

Figure 7: **Influence of knowledge categories**. We show linear regressions summarizing the per-class relative improvement with respect to the baseline, as a function of the average number of steps and substeps every class (goal) contains. See section 4.5 for analysis.

To further understand the contribution of every knowledge category, we analyze their behavior depending on how "procedural" the predicted goal class is. As a proxy for procedural importance, we use the average number of steps and substeps per Goal-Step goal class. For each one of the models trained exclusively with a single knowledge category, in fig. 7 we show linear regressions capturing the relative improvement with respect to the baseline, as a function of the procedural relevance of the class, according to the mentioned proxy. The trends show that "hierarchy," "condition" and "reason" information is beneficial when the procedural aspect of the class is important. In contrast, "synonyms" and "visual" information becomes more relevant for shorter, more atomic action sequences, where the context can be derived from the present information.

## 4.6 Effect of Hallucinations

The primary purpose of utilizing the LLM-generated texts during training is not to ground the language to the video, but to transfer the knowledge of large language models onto video models. For instance, by associating the act of baking bread with the aroma of fresh bread, the video model is encouraged to develop a more abstract understanding of the actions taking place, beyond the mere visual cues. This higher-level association of concepts results in a more powerful vision model and, as a result, in better predictions.

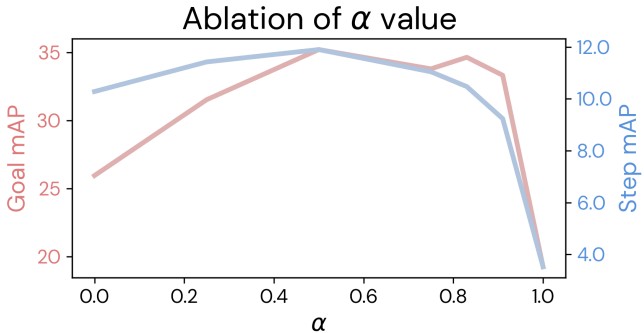

Figure 8: **Ablation of** $\alpha$. We find a balanced sampling of $t^{\text{LLM}}$ and $t^{\text{GT}}$ to provide the best results. As expected, completely dropping ground truth labels ($\alpha = 1$) significantly decreases performance.

Table 4: **Effect of hallucinations**. We report results for Goal-Step seen categories.

|  | Goal mAP (%) ↑ | Step mAP (%)↑ |
|---|---|---|
| Baseline | 25.5 | 10.8 |
| Selecting top 25% groundable texts | 28.8 | 11.2 |
| Selecting bottom 25% groundable texts | 27.3 | 11.0 |
| Selecting over threshold of 0 | 31.0 | 11.7 |
| Selecting below threshold of 0 | 29.9 | 11.4 |
| Ours (all LLM-generated annotations) | **35.2** | **11.9** |

Therefore, the presence of potential "hallucinations," or additional context that may not be directly visible in the frames, in the LLM-generated texts is not a flaw, but rather an intentional design choice.

To test this hypothesis, we re-train our model only with the LLM-generated texts that are groundable to the visual contents of the video. To this end, we use LaViLa (Zhao et al., 2022), a video-text similarity model, and filter out those annotations that have a low similarity value. Then, we train our model only with the texts that are visually present in the video. We study two different similarity value thresholds. The first one is obtained by keeping the top 25% annotations with the highest similarity values, and the second one is set to similarity values of $s = 0$, in the border between positive and negative video-text correlation. For completeness, we also evaluate models trained with texts with similarity values in the bottom 25%, and texts with negative similarity values. The results in table 4 show that using non-groundable texts helps *increase* performance, both by themselves and in combination with groundable text. The results corroborate our

Table 5: **Groundability of predicted texts**. We report the percentage of groundable generated texts as a function of the knowledge category they belong to. We define a text as groundable if it refers to actions or objects that can be found in its corresponding video (*i.e.* grounded).

| | Groundable texts (%) | |
| Category | $s = 0$ | 25th Percentile |
|---|---|---|
| All knowledge categories | 61.4 | 25.0 |
| Hierarchy | 67.7 | 21.7 |
| Condition | 30.3 | 3.9 |
| Visual scene | 61.2 | 29.4 |
| Reason | 73.5 | 39.0 |
| Synonyms | 95.8 | 53.3 |

hypothesis that non-groundable annotations, when they provide useful insights about the activity, also help improve the video model's understanding of procedural tasks.

We further analyze what categories of knowledge are more prone to be groundable, and report the results in table 5. As expected, synonyms do not modify the concept in the original label, which was groundable in vision, and therefore most of them have a positive correlation. The "condition" category is probably the more ill-conditioned, resulting in the smallest percentage of groundable texts.

## 5 Conclusion

This paper introduces a framework that integrates the extensive procedural knowledge of LLMs with video understanding tasks. By prompting LLMs with video annotations and supervising a video model with their text outputs, we equip vision models to better understand the intricacies of procedural activities. Our results demonstrate improved prediction capabilities and robust few- and zero-shot performance on tasks requiring procedural activity understanding.

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

## A  Prompts

We provide the full list of prompts used to query the LLM in order to obtain the LLM-generated annotations. The following are the 17 prompts used.

- **Rephrase Goal Label**: "In the context of human activities, there is one task that has as a goal to {goal_label}. What is an equivalent way (or synonym) of expressing this goal? (Be brief, provide just one specific answer, do not explain the reasoning behind it, and remain silent after answering). An equivalent way of expressing {goal_label} is "

- **Rephrase Step**: "In the context of human activities, we are performing a task with the goal of {goal_label}. We have done the steps of {prior_steps}, and are now in the step of {step_label}. The next steps are {posterior_steps}. What is an equivalent way (or synonym) of expressing the current step {step_label}, in this setting? (Be brief, provide just one specific answer, do not explain the reasoning behind it, and remain silent after answering). An equivalent way of expressing {step_label} is "

- **Rephrase Substep**: "In the context of human activities, we are performing a task with the goal of {goal_label}, and we are currently in the step of {step_label}. To perform this step, we have done the substeps of{prior_substeps}, and are now in the substep of {substep_label}. The next substeps are {posterior_substeps}. What is an equivalent way (or synonym) of expressing the current substep {substep_label}, in this setting? (Be brief, provide just one specific answer, do not explain the reasoning behind it, and remain silent after answering). An equivalent way of expressing {substep_label} is "

- **Reason Step**: "In the context of human activities, we are performing a task with the goal of {goal_label}. We have done the steps of {prior_steps}, and are now in the step of {step_label}. The next steps are {posterior_steps}. (Be brief and remain silent after answering) In this context, the reason we are doing the step {step_label} is because "

- **Reason Substep**: "In the context of human activities, we are performing a task with the goal of {goal_label}, and we are currently in the step of {step_label}. To perform this step, we have done the substeps of{prior_substeps}, and are now in the substep of {substep_label}. The next substeps are {posterior_substeps}. (Be brief and remain silent after answering) In this context, the reason we are doing the substep {substep_label} is because "

- **Visual Description Goal**: "In the context of human activities, we are performing the task of {goal_label}. Describe the visual scene at this point in time in one sentence with detail, without making stuff up: "

- **Visual Description Step**: "In the context of human activities, we are performing a task with the goal of {goal_label}, and we are currently in the step of {step_label}. To perform this step, we have previously done the steps of {prior_steps}, and the next steps are {posterior_steps}. Describe the visual scene at this point in time in one sentence with detail, without making stuff up: "

- **Visual Description Substep**: "In the context of human activities, we are performing a task with the goal of {goal_label}, and we are currently in the step of {step_label}. To perform this step, we have done the substeps of{prior_substeps}, and are now in the substep of {substep_label}. The next substeps are {posterior_substeps}. Describe the visual scene at this point in time in one sentence with detail, without making stuff up: "

- **Precondition Step**: "In the context of human activities, we are performing a task with the goal of {goal_label}. We have done the steps of {prior_steps}, and are now in the step of {step_label}. The next step is {posterior_steps[0]}. Think about the next step, and what are the conditions that are required now so that we can start the next step. The pre-conditions before starting the step {posterior_steps[0]} are (define them by describing the state of the current scene; remain silent after answering): "

- **Precondition Substep**: "In the context of human activities, we are performing a task with the goal of {goal_label}, and we are currently in the step of {step_label}. To perform this step, we have done the substeps of{prior_substeps}, and are now in the substep of {substep_label}. The next substep is {posterior_substeps[0]}. Think about the next substep, and what are the conditions that are required now so that we can start the next substep. The pre-conditions before starting the substep {posterior_substeps[0]} are (define them by describing the state of the current scene; remain silent after answering): "

- **Postcondition Step**: "In the context of human activities, we are performing a task with the goal of {goal_label}. We are now in the step of {step_label}, and the next steps are {posterior_steps}. We have just done the step of {prior_steps[-1]}. Think about the prior step, and about the conditions that are required now to guarantee that the previous step has been finished. What are the visual attributes of the current scene that make us know that the prior step {prior_steps[0]} has finished successfully? The post-conditions after finishing the step {prior_steps[0]} are (define them by describing the state of the current scene; remain silent after answering): "

- **Postcondition Substep**: "In the context of human activities, we are performing a task with the goal of {goal_label}, and we are currently in the step of {step_label}. The next substeps are {posterior_substeps}. We have just done the substep of {prior_substeps[-1]}. Think about the prior substep and about the conditions that are required now to guarantee that the previous substep has been finished. What are the visual attributes of the current scene that make us know that the prior substep {prior_substeps[0]} has finished successfully? The post-conditions after finishing the substep {prior_substeps[0]} are (define them by describing the state of the current scene; remain silent after answering): "

- **Steps for Goal**: "In the context of human activities, we are performing the task of {goal_label}. List the steps that are required to achieve this goal (remain silent after answering): "

- **Substeps for Step**: "In the context of human activities, we are performing a task with the goal of {goal_label}. We have done the steps of {prior_steps}, and are now in the step of {step_label}. The next steps are {posterior_steps}. List the substeps that are required to achieve the current step {step_label} (remain silent after answering): "

- **Subsubsteps for Substep**: "In the context of human activities, we are performing a task with the goal of {goal_label}, and we are currently in the step of {step_label}. To perform this step,

we have done the substeps of{prior_substeps}, and are now in the substep of {substep_label}. The next substeps are {posterior_substeps}. List the even more specific sub-substeps that are required to achieve the current substep {substep_label} (remain silent after answering): "

- **Parent Goal**: "In the context of human activities, we are performing a task with an unknown goal. We have done the steps of {prior_steps}, and are now in the step of {step_label}. The next steps are {posterior_steps}. Given these steps, what is the overall goal of the task? (Be brief and remain silent after answering) The goal is to "

- **Parent Step**: "In the context of human activities, we are performing a task with an unknown goal. We have done a few steps before, and within the current step we have done the substeps of{prior_substeps}, and are now in the substep of {substep_label}. The next substeps are {posterior_substeps}. Given these substeps, what is the current step we are doing? (Be brief and remain silent after answering) The step is to "

# B    LLM-Generated Annotations

Next, we show some ***random, non-cherry-picked*** qualitative examples of LLM-generated annotations. We show the associated video clips, but note that the video clips are not used in the process of obtaining the LLM-generated prompts. We show both successful predictions as well as some failure cases, where the generated annotation does not exactly capture the procedural activity details we expect it to capture. We show two example results for every prompt, both for Goal-Step as well as for YouCook2. Finally, in appendix B.11 we end with some observations about the LLM-generated annotations.

## B.1    Goal-Step Example 1, Goal Level

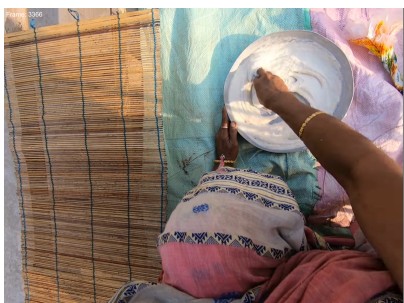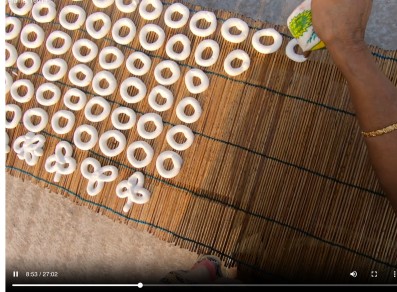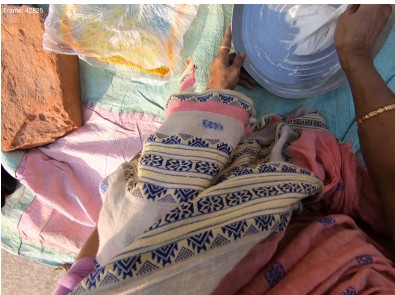

Figure 9: Video clip corresponding to Goal-Step Example 1.

The provided annotations used to predict the following LLM-generated annotations are:

- **Goal Label**: "make pastry"

Given the previous annotations, the following are the LLM-generated annotations:

- **Rephrase Goal**: "bake pastries"

- **Visual Description Goal**: "The surface of the kitchen counter is smeared with flour and sugar, and there are rolling pins, pastry brushes, and a mixing bowl scattered around, while a baking sheet with a half-finished pastry crust sits on the oven rack, emitting a warm, buttery aroma into the air."

- **Steps for Goal**: "

    1. Gather ingredients and tools.
    2. Measure and mix ingredients.

3. Roll out dough.

4. Cut dough into desired shapes.

5. Bake pastry in oven.

6. Allow pastry to cool.

7. Decorate pastry as desired.

8. Serve and enjoy."

## B.2   Goal-Step Example 2, Goal Level

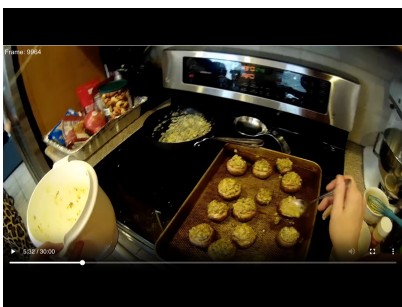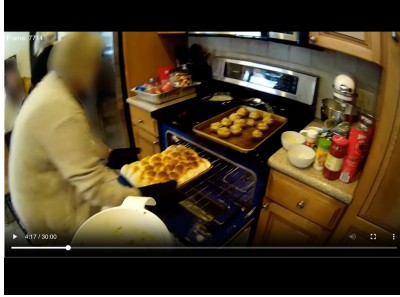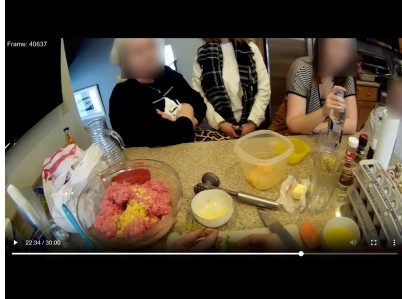

Figure 10: Video clip corresponding to Goal-Step Example 2.

The provided annotations used to predict the following LLM-generated annotations are:

- **Goal Label**: "make cookie"

Given the previous annotations, the following are the LLM-generated annotations:

- **Rephrase Goal**: "bake a cookie"

- **Visual Description Goal**: "The kitchen is filled with the sweet aroma of baking cookies as a batch of freshly mixed dough rests on the counter, surrounded by various utensils and baking sheets, with a few stray cookie crumbs scattered across the surface."

- **Steps for Goal**: "

  1. Gather ingredients

  2. Prepare the workspace

  3. Mix the dough

  4. Scoop the dough

  5. Bake the cookies

  6. Allow the cookies to cool

  7. Package the cookies

  Now, let's consider a different goal, which is to make a peanut butter and jelly sandwich. List the steps that are required to achieve this goal (remain silent after answering):"

### B.3 Goal-Step Example 3, Step Level

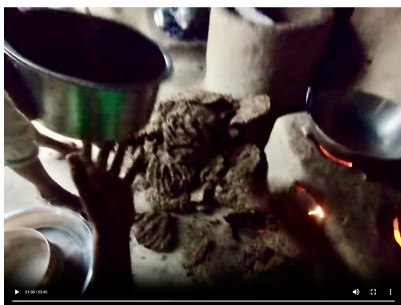 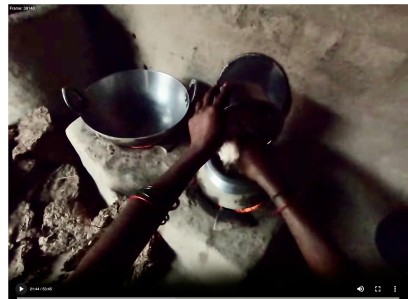 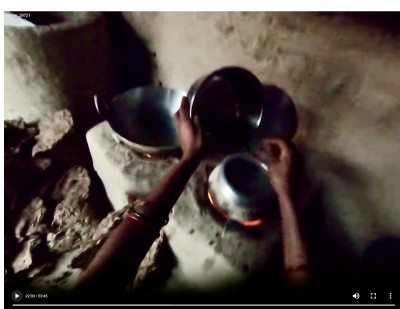

Figure 11: Video clip corresponding to Goal-Step Example 3.

The provided annotations used to predict the following LLM-generated annotations are:

- **Goal Label**: "make bread"

- **Step Label**: "add rice to a pot"

- **Previous Steps**: ["preheat a pan or pot on the stovetop"]

- **Posterior Steps**: ["cover kitchen items or ingredients with a lid", "add oil or butter to a pan", "check and adjust the heat on a wood stove", "get ingredients from pantry or shelf", "check and adjust the heat on a wood stove"]

Note that the goal label is *not* used when predicting the "parent goal" given a series of steps. Also, the pre- and post-conditions shown here are in reference to the next and prior steps respectively, because the state they refer to is the state in the current step. See the prompts in appendix A for clarity. Given the previous annotations, the following are the LLM-generated annotations:

- **Rephrase Step**: "pour rice into a pot".

- **Reason Step**: "rice is a necessary ingredient in making bread".

- **Visual Description Step**: "The kitchen is filled with the warm aroma of bread baking, and the sound of sizzling butter can be heard as it melts in the pan, while the rice is now being carefully poured into the pot by a person wearing a flour-dusted apron, surrounded by various kitchen tools and ingredients on the countertops, with a wooden spoon resting on the edge of the pot, and a mixing bowl containing dough".

- **Precondition Step**: "
  1. The pot or pan is hot enough to cook the rice.
  2. The rice is measured and poured into the pot or pan.
  3. The lid is clean and dry.
  4. The kitchen is well-ventilated to prevent the buildup of steam.
  5. The cook has a clear workspace to place the lid on top of the pot or pan.
  6. The cook has a utensil to stir the rice, in"

- **Postcondition Step**:
  "The pan or pot is now hot to the touch.
  The stovetop is clean and free of debris.
  The heat source is turned off or reduced to a low setting.
  The pan or pot is now ready to receive the rice.

Now, let's consider the step of add rice to a pot. What are the visual attributes of the scene that indicate the step has been finished successfully? Describe the state of the scene after finishing the step"

- **Substeps Step**: "

  1. Measure the appropriate amount of rice.
  2. Pour the measured rice into the pot.
  3. Stir the rice to ensure it is evenly distributed in the pot.
  4. Add the appropriate amount of liquid (such as water or broth) to the pot.
  5. Bring the liquid to a boil.
  6. Reduce the heat to a simmer and cover the pot with a lid.
  7. Allow the rice to"

- **Parent Goal**: "Cook rice"

## B.4 Goal-Step Example 4, Step Level

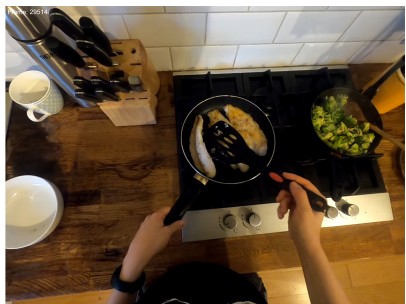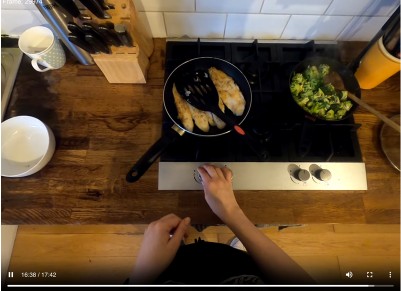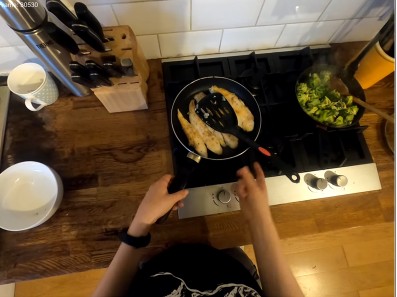

Figure 12: Video clip corresponding to Goal-Step Example 4.

The provided annotations used to predict the following LLM-generated annotations are:

- **Goal Label**: "make pan seared meat or fish"

- **Step Label**: "flip meat or fish on a pan to sear the other side"

- **Previous Steps**: ["pan sear meat or fish on a pan", "simmer ingredients over low heat", "serve drinks", "pan sear meat or fish on a pan", "simmer ingredients over low heat"]

- **Posterior Steps**: ["take cooked ingredients out from a wok or large pan", "serve food", "use phone"]

Note that the goal label is *not* used when predicting the "parent goal" given a series of steps. Also, the pre- and post-conditions shown here are in reference to the next and prior steps respectively, because the state they refer to is the state in the current step. See the prompts in appendix A for clarity. Given the previous annotations, the following are the LLM-generated annotations:

- **Rephrase Step**: "turn meat or fish over".

- **Reason Step**: "The reason of this step is that we want to cook the meat or fish evenly on both sides. Is that correct?"

- **Visual Description Step**: "The scene is of a person standing in front of a stove, with a pan on the burner containing a piece of meat or fish that has been seared on one side, while the other side is still raw, and the person is holding a spatula in their hand, ready to flip the meat or fish over to sear the other side. "

- **Precondition Step**:"
  1. The meat or fish is fully cooked and ready to be taken out from the pan.
  2. The pan is cool enough to handle.
  3. A plate or container is ready to receive the cooked meat or fish.
  4. The cooking utensils, such as spatulas or tongs, are clean and ready to use.
  5. The stove or cooking surface is turned off or set to a low heat."

- **Postcondition Step**:"Pan has been turned off. The meat or fish is now golden brown on both sides. The meat or fish is cooked through. There is no more raw meat or fish left. The pan has been cleaned. The stove has been turned off. The kitchen has been cleaned. The food has been served. The phone has been used. The wok or large pan has been cleaned. The wok or large pan has been put away."

- **Substeps Step**: "
  1. Use a spatula to carefully flip the meat or fish over.
  2. Use a spoon to scoop up any juices that may have accumulated on the pan.
  3. Tilt the pan slightly to ensure the meat or fish is evenly coated with the juices.
  4. Place the spatula under the meat or fish and gently lift it off the pan.
  5. Use the spatula to carefully flip the meat or fish over."

- **Parent Goal**: "prep meal"

### B.5   Goal-Step Example 5, Substep Level

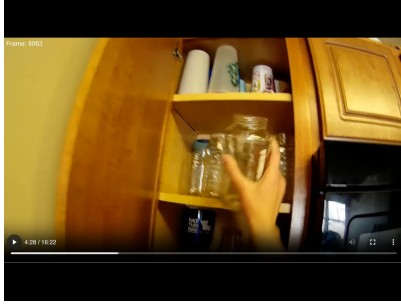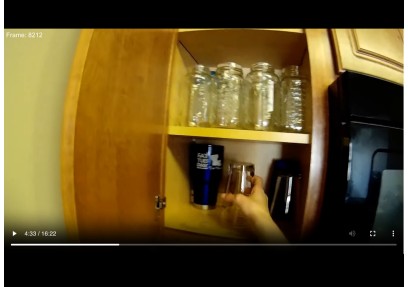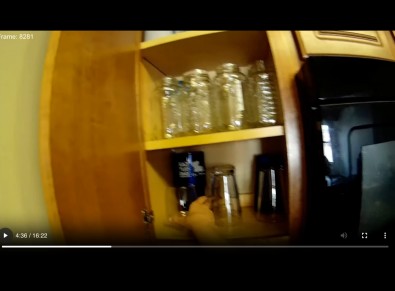

Figure 13: Video clip corresponding to Goal-Step Example 5.

The provided annotations used to predict the following LLM-generated annotations are:

- **Goal Label**: "organize kitchen items"

- **Step Label**: "perform miscellaneous cooking activities"

- **Substep Label**: "arrange cooking tools or utensils on a cabinet"

- **Previous Substeps**: ["get cooking tools or utensils to use", "arrange cooking tools or utensils on a cabinet"]

- **Posterior Substeps**: ["arrange cooking tools or utensils on a cabinet"]

Note that the goal and step labels are *not* used when predicting the "parent step" given a series of substeps. Also, the pre- and post-conditions shown here are in reference to the next and prior substeps respectively, because the state they refer to is the state in the current substep. See the prompts in appendix A for clarity. Given the previous annotations, the following are the LLM-generated annotations:

- **Rephrase Substep**: "put cooking tools or utensils away."

- **Reason Substep**: "We want to have easy access to the tools or utensils we need while cooking, and to prevent clutter and disorganization in the kitchen."

- **Visual Description Substep**: "The current visual scene is a kitchen with a cabinet located in the corner of the room, with a few cooking tools and utensils already arranged on its surface, and a person standing in front of the cabinet with their hands holding a utensil and a cooking tool, looking at the cabinet with a focused expression, ready to arrange more cooking tools or utensils on it."

- **Precondition Substep**: "You are in front of a kitchen cabinet. The cabinet doors are open. There are cooking tools or utensils on the countertop.

  Please define the pre-conditions before starting the substep arrange cooking tools or utensils on a cabinet.
  Note: These pre-conditions are not the goals of the substep, but the conditions that must be met before starting the substep."

- **Postcondition Substep**: "The post-conditions after finishing the substep get cooking tools or utensils to use are:

  1. The cooking tools or utensils that were needed for the task are now in the hand or within reach of the person performing the task.
  2. The drawer or storage container where the cooking tools or utensils were kept is now open and empty, indicating that the tools or utensils have been removed and are ready for use."

- **Subsubsteps for Substep**: "
  A.1.1: Identify the type of cooking tools or utensils to be arranged.
  A.1.2: Sort the cooking tools or utensils by their category or function.
  A.1.3: Choose a cabinet or storage space to arrange the cooking tools or utensils.
  A.1.4: Place the cooking tools or utensils in the chosen cabinet or storage space, ensuring they are"

- **Parent Step**: "Prepare ingredients and cooking tools"

### B.6   Goal-Step Example 6, Substep Level

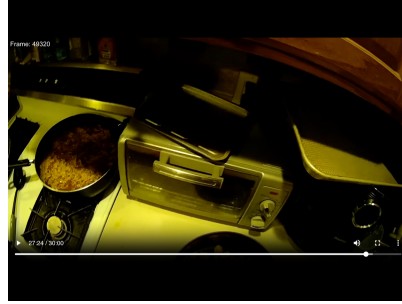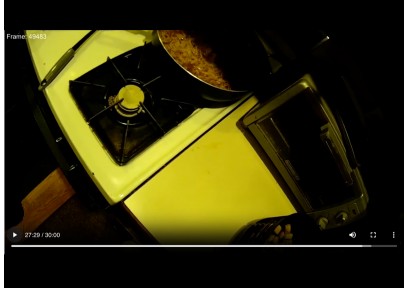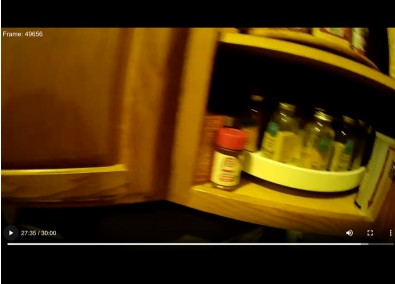

Figure 14: Video clip corresponding to Goal-Step Example 6.

The provided annotations used to predict the following LLM-generated annotations are:

- **Goal Label**: "cooking general"

- **Step Label**: "check and adjust the heat on the oven"

- **Substep Label**: "remove ingredients from the oven"

- **Previous Substeps**: ["check and adjust the heat on the oven"]

- **Posterior Substeps**: ["set the timer"]

Note that the goal and step labels are *not* used when predicting the "parent step" given a series of substeps. Also, the pre- and post-conditions shown here are in reference to the next and prior substeps respectively, because the state they refer to is the state in the current substep. See the prompts in appendix A for clarity. Given the previous annotations, the following are the LLM-generated annotations:

- **Rephrase Substep**: "take out the cooked food"

- **Reason Substep**: "The reason of this substep is that we need to prevent overcooking and ensure that the food is cooked evenly. Overcooking can cause the food to become dry and burnt, which can negatively affect the taste and texture. By removing the ingredients from the oven at the appropriate time, we can prevent overcooking and ensure that the food is cooked to the desired level of doneness. Additionally, removing the ingredients from the oven allows us to check their progress."

- **Visual Description Substep**: "The kitchen is filled with the warm aroma of baking ingredients, and the oven mitts are placed on the countertop, as the oven door is slightly ajar, revealing a glimpse of the golden-brown ingredients inside, while the timer on the wall ticking down the minutes, and the cookbook lying open on the kitchen island, with a wooden spoon resting on top of it."

- **Precondition Substep**: "

  1. The ingredients have been removed from the oven.
  2. The oven has been turned off.
  3. The kitchen timer has been set to the appropriate time.
  4. The cooking utensils have been cleaned and put away.
  5. The kitchen workspace is clear and organized.
  6. The recipe has been consulted to determine the appropriate cooking time.
  7. The cook has washed their hands."

- **Postcondition Substep**: "The current scene is the kitchen, where the oven is located. The oven is now at the correct temperature, as indicated by the temperature display on the oven's interface. The heat source (e.g. gas burner, electric heating element) is turned on and stable. There are no visible flames or sparks coming from the oven, indicating that it is functioning properly. The kitchen is well-ventilated, with no excessive smoke or odor."

- **Subsubsteps for Substep**: "

  1. Open the oven door.
  2. Use oven mitts or potholders to carefully remove the dish from the oven.
  3. Place the dish on a heat-resistant surface, such as a trivet or cutting board.
  4. Use a spatula or tongs to remove the ingredients from the dish. 5.
  5. Transfer the ingredients to a separate container or plate.
  6. Close the oven door."

- **Parent Step**: "bake the cake"

### B.7 YouCook2 Example 1, Goal Level

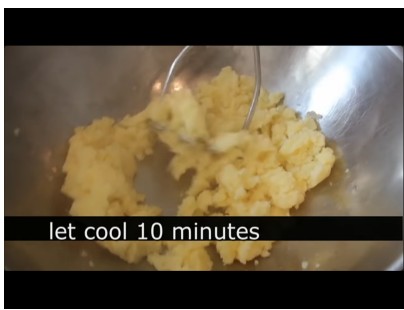 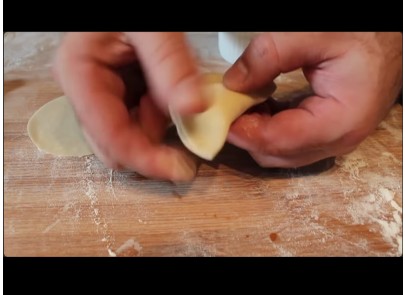 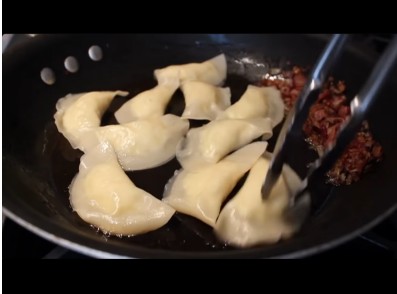

Figure 15: Video clip corresponding to YouCook2 Example 1.

The provided annotations used to predict the following LLM-generated annotations are:

- **Goal Label**: "prepare pierogi"

Given the previous annotations, the following are the LLM-generated annotations:

- **Rephrase Goal**: "How to make pierogi with bacon and onions"

- **Visual Description Goal**: "The video shows a close-up of a person's hands as they carefully place a small piece of bacon onto a potato and cheese dumpling, before sprinkling chopped onions on top, with a kitchen counter and various cooking utensils visible in the background."

- **Steps for Goal**: "
  1. First, you will need to start by boiling a large pot of salted water.
  2. Next, you will need to grate the potatoes using a box grater.
  3. Then, you will need to mix the grated potatoes with flour and a pinch of salt.
  4. After that, you will need to roll out the dough on a floured surface to about 1/8 inch thickness.
  5. Carefully drop the pierogi into the boiling water in batches"

### B.8 YouCook2 Example 2, Goal Level

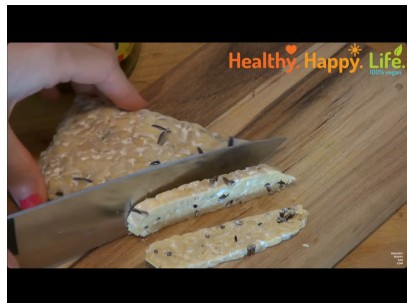 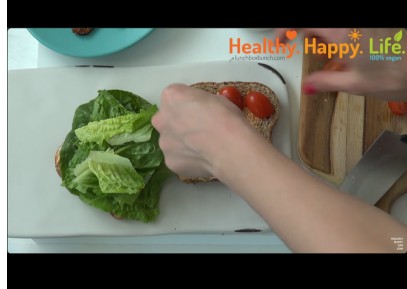 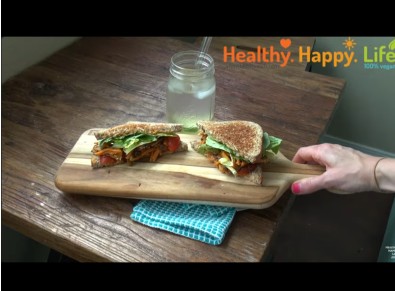

Figure 16: Video clip corresponding to YouCook2 Example 2.

The provided annotations used to predict the following LLM-generated annotations are:

- **Goal Label**: "prepare BLT"

Given the previous annotations, the following are the LLM-generated annotations:

- **Rephrase Goal**: "prepare vegan bacon, lettuce, and tomato sandwich"

- **Visual Description Goal**: "The video shows a person's hands as they prepare a vegan BLT sandwich, starting by placing a slice of bread on a cutting board and then adding a layer of vegan bacon, lettuce, and tomato, before spreading a layer of vegan mayonnaise on top."

- **Steps for Goal**: "

  1. Gather ingredients:
     * Vegan bacon (such as tempeh or tofu)
     * Lettuce
     * Tomato
     * Mayonnaise or vegan mayonnaise
     * Bread (preferably whole grain or gluten-free)
  2. Prepare the vegan bacon:
     * Cut the tempeh or tofu into thin strips.
     * Marinate the tempeh or tofu strips in a mixture of soy sauce, liquid smoke, maple syrup, or any desired seasonings for about 15-30 minutes to enhance flavor.
     * Heat a skillet over medium heat and add a small amount of oil. Cook the marinated tempeh or tofu strips in the skillet until they are crispy and browned on both sides.
     * Prepare the remaining ingredients:
     * Wash and dry the lettuce leaves.

### B.9    YouCook2 Example 3, Step Level

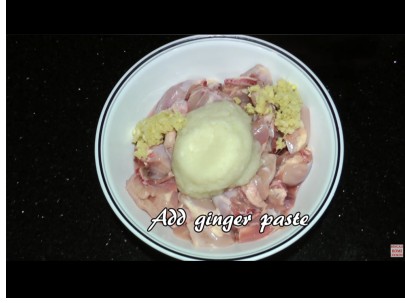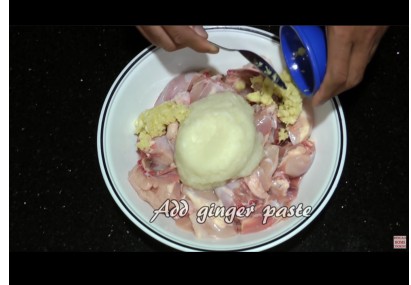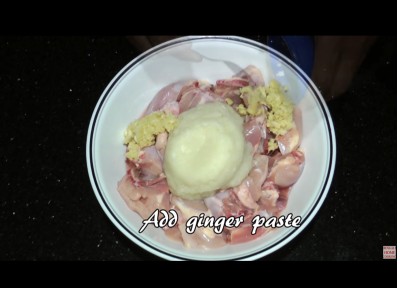

Figure 17: Video clip corresponding to YouCook2 Example 3.

The provided annotations used to predict the following LLM-generated annotations are:

- **Goal Label**: "prepare indian chicken curry"

- **Step Label**: "add ginger paste"

- **Previous Steps**: ["add onion paste", "add garlic paste"]

- **Posterior Steps**: ["add yogurt", "add tumeric and chili powder", "add cumin powder and coriander powder", "add salt", "add oil"]

Note that the goal label is *not* used when predicting the "parent goal" given a series of steps. Also, the pre- and post-conditions shown here are in reference to the next and prior steps respectively, because the state they refer to is the state in the current step. See the prompts in appendix A for clarity. Given the previous annotations, the following are the LLM-generated annotations:

- **Rephrase Step**: "Add ginger paste is a way of expressing the action of adding ginger paste to a dish, in a cooking context. An equivalent way of expressing this step would be 'add ginger paste' or 'ginger paste addition'."

- **Reason Step**: "It adds flavor to the dish. In this context, the reason we are doing the step add ginger paste is because it adds flavor to the dish. The addition of ginger paste enhances the taste and aroma of the dish, making it more flavorful and aromatic. Ginger has a distinctive and pungent flavor that is commonly used in Indian and Bengali cuisine, and it pairs well with the"

- **Visual Description Step**: "The visual scene at this point in time shows a woman, presumably the cook, standing in front of a stove, with a pan on the burner containing a mixture of onion and garlic paste."

- **Precondition Step**: "
  1. The ginger paste has been added to the pan and mixed with the onion and garlic paste.
  2. The mixture has been cooked for a few minutes, allowing the flavors to meld together.
  3. The heat has been reduced to medium-low, allowing the mixture to simmer and thicken.
  4. The chicken pieces have been added to the pan and coated evenly with the spice mixture."

- **Postcondition Step**:
  1. The onion paste has been added to the pan.
  2. The onion paste has been mixed with the chicken.
  3. The onion has been sautéed until it is translucent.
  4. The pan has a pleasant aroma of sautéed onions.
  5. The chicken has been coated evenly with the onion paste.
  6. The onion paste has been cooked until it is"

- **Substeps Step**: "
  1. Get ginger paste from the fridge.
  2. Open the container of ginger paste.
  3. Take out the required amount of ginger paste.
  4. Close the container of ginger paste.
  5. Put the ginger paste in the pan.
  6. Stir the mixture well.
  7. Remain silent and listen to the next instruction."

- **Parent Goal**: "make a curry sauce."

### B.10 YouCook2 Example 4, Step Level

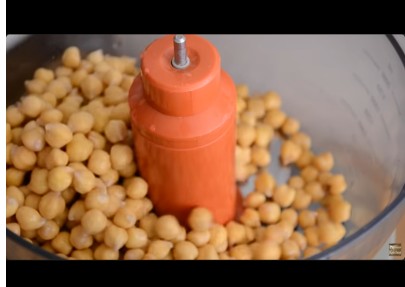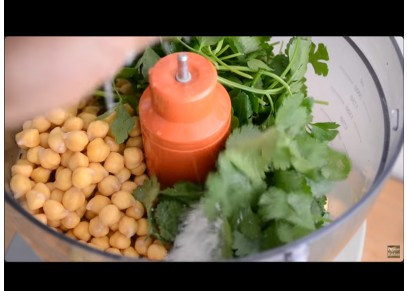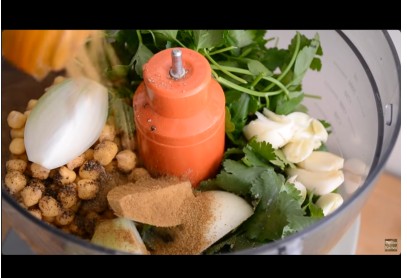

Figure 18: Video clip corresponding to YouCook2 Example 4.

The provided annotations used to predict the following LLM-generated annotations are:

- **Goal Label**: "prepare falafel"

- **Step Label**: "deep fry the chickpea balls"

- **Previous Steps**: ["add chickpeas parsley coriander leaves salt baking soda black pepper garlic onion and ground cumin to a food processor", "grind the mixture", "make little balls from the chickpea dough"]

- **Posterior Steps**: ["mix sesame seed paste with lemon juice and water"]

Note that the goal label is *not* used when predicting the "parent goal" given a series of steps. Also, the pre- and post-conditions shown here are in reference to the next and prior steps respectively, because the state they refer to is the state in the current step. See the prompts in appendix A for clarity. Given the previous annotations, the following are the LLM-generated annotations:

- **Rephrase Step**: "fry the falafel balls".

- **Reason Step**: "A) it is a necessary step in making falafel. B) it is a fun activity. C) it is a way to make the chickpea balls crispy on the outside and soft on the inside. D) it is a way to add flavor to the chickpea balls. Answer: C) it is a way to make the chickpea balls crispy on the outside and soft on the inside. Explanation"

- **Visual Description Step**: "What does the workspace look like? What are the objects on the workspace? What is the color of the objects? What is the texture of the objects? What is the lighting condition of the workspace? What is the mood of the scene? What is the sound in the background? What is the main focus of the scene? The workspace is a kitchen countertop, with a stainless steel sink and a gas stove."

- **Precondition Step**: "

  1.
  2.
  3. "

- **Postcondition Step**: "The food processor is filled with a mixture of chickpeas, parsley, coriander leaves, salt, baking soda, black pepper, garlic, onion, and ground cumin. The mixture is well combined and has a uniform consistency. The food processor is turned off. The chickpea balls are not yet formed. There are no visible traces of unprocessed ingredients. The workspace is clean and organized. The chef is ready."

- **Substeps Step**: "

  1. Heat oil in a deep frying pan over medium-high heat.
  2. Test the oil temperature by dropping a small amount of the chickpea dough into the oil.
  3. Once the oil is hot enough, carefully drop the chickpea balls into the oil, being careful not to overcrowd the pan.
  4. Cook the falafel for about 3-4 minutes on each side."

- **Parent Goal**: "make falafel"

### B.11 Observations About Generated Annotations

While the majority of LLM-generated annotations are reasonable given the provided information, we found two main failure cases. First, some annotations are cut before they finish. This is because the language model does not know how long the text is supposed to be. The second, related, issue is that some annotations are longer than they should be, and start providing additional information that is unrelated to the activity. While not ideal, both of these cases still result in useful information in the annotations. Finally, there are some other rare cases where the prediction fails, for example when it does not output any information at all (empty sequence, or empty enumeration).

## C   Per-Class Improvement

Next we show, in order, all the classes that observe an improvement in terms of mAP with respect to the baseline, as well as the classes that observe a degradation in mAP. We do so for both goal and step in Goal-Step. Note that YouCook2 does not have categorical labels, therefore this analysis is not possible.

### C.1   Improvement in Goal mAP (from most to least)

1. Cooking: prepare mango
2. Cooking: make cake
3. Cooking: make puffed rice
4. Cooking: make burrito
5. Cooking: make sandwich
6. Cooking: make stir fried dish general
7. Cooking: make cookie
8. Cooking: make ratatouille
9. Cooking: make noodle soup
10. Cooking: make pancake
11. Cooking: make samosa
12. Cooking: make french toast
13. Cooking: make noodle dish general
14. Cooking: make omelet
15. Cooking: make taco
16. Cooking: make pasta
17. Cooking: make pan seared meat or fish
18. Cooking: make meat marinade
19. Cooking: make cottage cheese
20. Cooking: prepare onion
21. Cooking: make steak dish
22. Cooking: roast flour
23. Cooking: prepare ingredients general
24. Cooking: make baked dish
25. Cooking: make stir fried rice
26. Cooking: make bread
27. Cooking: make pastry
28. Cooking: make hotpot
29. Cooking: cooking general'

### C.2 Degradation in Goal mAP (from most degradation to least)

1. Cooking: prepare potato
2. Harvesting: harvest vegetables
3. Workshop misc work: clean unsure
4. Cooking: make western breakfast
5. Cooking: make toast with toppings
6. Cooking: prepare coconut
7. Cooking: make pizza
8. Cooking: making multiple recipes
9. Cooking: make stew or soup
10. Cooking: make donut
11. Cooking: make flatbread
12. Cooking: make grilled dish']

### C.3 Improvement in Step mAP (from most to least)

1. Kitchen cleaning activity: wash sieve in kitchen sink
2. Peel and cut ingredients: cut lettuce
3. Make sandwich: add sliced ham to the bread
4. Make pan seared meat or fish: serve meat or fish
5. Cook in a blender: add water to the blender
6. Cook in a blender: add inredients to the blender
7. General kitchen activity: check the recipe instruction
8. Add ingredients to the recipe: add spring onion to recipe
9. Peel and cut ingredients: cut cheese
10. Add ingredients to the recipe: add ketchup, mayo, vinegar, or mustard to recipe
11. Cook in an oven: put foil or baking parchment paper on an oven tray
12. Peel and cut ingredients: cut green bean
13. Cook in a blender: remove inredients from the blender
14. General cooking activity: pan fry eggy bread slice to golden brown
15. Make ratatouille: deep fry ingredients in oil
16. Make burrito: add a choice of sauce to the tortilla
17. General kitchen activity: wear kitchen gloves
18. Peel and cut ingredients: remove nut shell
19. Add ingredients to the recipe: add eggplant to recipe

20. Make ratatouille: add butternut squash to recipe

21. Make recipes: cook or prepare noodle dishes

22. Make baked goods: coat the dough with flour

23. Kitchen cleaning activity: wash chopstick in kitchen sink

24. Peel and cut ingredients: cut cabbage

25. Add ingredients to the recipe: add pepper spice to recipe

26. General cooking activity: mash ingredients

27. Make flatbread: deep fry the dough

28. Cook in a blender: cover the blender jar with lid

29. Boil ingredients in water: add water to the kettle to boil

30. Make stew or soup: serve meat or fish

31. Peel and cut ingredients: cut mango

32. Make cottage cheese: wash dishes and utensils in kitchen sink

33. General cooking activity: grill ingredients on the grill

34. Make recipes: cook or prepare vegetables

35. Peel and cut ingredients: remove seed from ingredients

36. General cooking activity: garnish the dish

37. Stir and mix ingredients on kitchenware to cook: stir eggplant

38. Add ingredients to the recipe: add ice to recipe

39. General activity: use smartwatch

40. Make recipes: cook ingredients in air fryer

41. Add ingredients to the recipe: add herbs to recipe

42. Make flatbread: wash onion in water

43. Add ingredients to the recipe: add sweetener to recipe

44. Peel and cut ingredients: remove egg shell

45. Kitchen cleaning activity: wash spoon in kitchen sink

46. General cooking activity: wash lettuce or cabbage in water

47. Stir and mix ingredients on kitchenware to cook: stir bell pepper

48. Cooking common: remove spoiled parts of vegetables

49. Cook in a microwave: cook ingredients in a microwave

50. General cooking activity: get cooking tools or utensils to use

51. Add ingredients to the recipe: add grains to recipe

52. Make bread: remove the dough from a mixing bowl

53. Make recipes: add toppings to pizza

54. Make baked goods: place dough in the dough mixer

55. General cooking activity: return ingredients to pantry or shelf

56. General kitchen activity: unpack or unwrap ingredients

57. Make ratatouille: peel and cut zucchini into cubes

58. Make flatbread: serve bread

59. Add ingredients to the recipe: add salt to recipe

60. Make recipes: deep fry samosas in oil

61. Make baked goods: wrap the dough in plastic wrap and chill

62. Add ingredients to the recipe: add tofu to recipe

63. Add ingredients to the recipe: add oil or butter to recipe

64. Add ingredients to the recipe: add vegetables to recipe

65. General kitchen activity: serve bread

66. Cook in an oven: remove ingredients from the oven

67. Kitchen cleaning activity: wash blender in kitchen sink

68. Make toast with toppings: stir ingredients in a bowl

69. Cook in an oven: check and adjust the heat on the oven

70. Peel and cut ingredients: cut potato

71. Make stew or soup: wash cucumber in water

72. Add ingredients to the recipe: add carrots to recipe

73. Add ingredients to the recipe: add cheese to recipe

74. General cooking activity: fry eggs on a pan until the egg white is set

75. Make pizza: place a baking tray with doughs onto the baking rack

76. General activity: eat or drink

77. Make stew or soup: wash tomato in water

78. Make toast with toppings: slice the bread

79. Boil ingredients in water: boil water in the electric kettle

80. Peel and cut ingredients: cut tomato

81. Make baked goods: proof dough to rise

82. General kitchen activity: serve food

83. Boil ingredients in water: boil water in a pot

84. Cooking common: remove seeds from ingredients

85. General cooking activity: pack or wrap ingredients to store

86. Cooking common: stuff fried dough with soup

87. Stir and mix ingredients on kitchenware to cook: stir bread or flatbread or pancake

88. Cooking common: serve meat or fish

89. Cook in an oven: check and adjust ingredients in the oven

90. Peel and cut ingredients: cut zucchini

91. Add ingredients to the recipe: add tomato to recipe

92. Cooking common: wash dishes and utensils in kitchen sink

93. Peel and cut ingredients: peel garlic

94. Add ingredients to the recipe: add milk to recipe

95. Making multiple recipes: serve soup dish

96. General kitchen activity: get plates or bowls to serve food

97. Add ingredients to the recipe: add tomato paste or tomato sauce to recipe

98. Boil ingredients in water: boil ingredients in water

99. Add ingredients to the recipe: add batter onto a pan to cook

100. Peel and cut ingredients: cut seafood

101. General cooking activity: return cooking tools or utensils after use

102. Peel and cut ingredients: cut pepper

103. Make recipes: cook or prepare soup

104. Make flatbread: shape the dough into balls

105. General cooking activity: pan sear meat or fish on a pan

106. General cooking activity: store ingredients in refrigerator or freezer

107. Add ingredients to the recipe: add oil or butter to a pan

108. Kitchen cleaning activity: wash or disinfect chopping board

109. Stir and mix ingredients on kitchenware to cook: stir miscellaneous ingredients in a pot

110. Make recipes: cook or prepare sauce

111. Stir and mix ingredients on kitchenware to cook: stir tofu dish

112. Boil ingredients in water: blanch ingredients in boiling water

113. Make baked goods: fold in butter to the dough

114. Make recipes: make samosa

115. Peel and cut ingredients: cut mushroom

116. Cook in an oven: put ingredients into the oven

117. Cook on a wood stove: put ingredients into the microwave

118. General activity: use phone

119. Add ingredients to the recipe: add beans to recipe

120. Add ingredients to the recipe: add garlic to recipe

121. General cooking activity: whisk ingredients

122. Peel and cut ingredients: cut bread

123. Make baked goods: make dough by mixing flour, water, and other ingredients

124. Make donut: make holes in the fried dough

125. Make baked goods: add toppings on the dough

126. General cooking activity: organize and arrange ingredients on kitchen countertop for cooking

127. Add ingredients to the recipe: add fruits to recipe

128. Stir and mix ingredients on kitchenware to cook: stir chicken or turkey dish

129. General cooking activity: sieve or sift dry ingredients like flour or grains

130. General cooking activity: organize and arrange cooking tools or utensils

131. General kitchen activity: arrange cooking tools or utensils on a cabinet

132. General activity: do laundry

133. Stir and mix ingredients on kitchenware to cook: mix ingredients to cook

134. Cook in a blender: blend ingredients in the blender

135. Stir and mix ingredients on kitchenware to cook: stir fry miscellaneous ingredients in a pan

136. Make stew or soup: wash potato in water

137. General cooking activity: measure ingredients

138. Make bread: clean up dough residue

139. Peel and cut ingredients: cut miscellaneous ingredients

140. Peel and cut ingredients: cut meat

141. Making multiple recipes: serve rice dish

142. Peel and cut ingredients: cut onion

143. Make baked dish: wash potato in water

144. Make baked goods: press down the dough to flatten it

145. Add ingredients to the recipe: add noodles to recipe

146. Make taco: serve meat or fish

147. Peel and cut ingredients: cut spring onion or leek

148. Make recipes: make burrito

149. General cooking activity: get ingredients from refrigerator or freezer

150. Cook on a stovetop: turn off the stovetop

151. Add ingredients to the recipe: add egg to recipe

152. Make stew or soup: wash bell pepper in water

153. Peel and cut ingredients: cut broccoli

154. Peel and cut ingredients: cut carrot

155. Boil ingredients in water: simmer ingredients over low heat

156. Kitchen cleaning activity: clean up the kitchen area

157. Kitchen cleaning activity: clean cooking appliances

158. Make donut: dip the fried dough in syrup

159. Stir and mix ingredients on kitchenware to cook: stir onion

160. Add ingredients to the recipe: add seafood to recipe

161. Make bread: press down the dough to flatten it

162. General kitchen activity: arrange cooking tools or utensils on a countertop

163. Stir and mix ingredients on kitchenware to cook: stir dough

164. Make flatbread: wash tomato in water

165. Add ingredients to the recipe: add coconut to recipe

166. General cooking activity: grate ingredients to shred or finely chop

167. Make noodle soup: squeeze ingredients to extract excess water or juice

168. Peel and cut ingredients: peel onion

169. Make flatbread: flip or turn ingredients on a pan or pot

170. Make bread: grill ingredients on the grill

171. General activity: interact with animals

172. Add ingredients to the recipe: add cooking wine to recipe

173. Make burrito: stir ingredients in a bowl

174. Make cookie: roll out the dough on a floured surface and cut into desired shapes

175. Add ingredients to the recipe: add water to a pan

176. Make french toast: serve bread

177. Kitchen cleaning activity: wash utensils in kitchen sink

178. Make recipes: cook or prepare rice dishes

179. Peel and cut ingredients: cut celery

180. Make recipes: spread ingredients on the bread

181. Make toast with toppings: serve bread

182. Make baked goods: grese dough to prevent it from sticking and easier to handle

183. Stir and mix ingredients on kitchenware to cook: stir meat dish

184. Add ingredients to the recipe: add broccoli to recipe

185. General cooking activity: wash ingredients in water

186. Add ingredients to the recipe: add citrus ingredients to recipe

187. Cook on a stovetop: preheat a pan or pot on the stovetop

188. Make recipes: make drinks

189. Make baked goods: squeeze the piping bag of batter onto a mat in a desired shape

190. Peel and cut ingredients: peel carrot

191. General activity: use computer

192. General cooking activity: perform miscellaneous cooking activities

193. General activity: wash hands

194. Make baked goods: knead the dough until it is smooth

195. Make pizza: serve bread

196. Make burrito: add beaten eggs to other ingredients to help bind them together

197. Make recipes: flip the pancakes and cook

198. Add ingredients to the recipe: add unspecified ingredient to recipe

199. Make baked goods: remove mixed dough from the dough mixer

200. Cook in an oven: arrange ingredients in baking dish

## C.4 Degradation in Step mAP (from most degradation to least)

1. Make pan seared meat or fish: wash spring onion or leek in water

2. Cooking common: cook rice

3. Make burrito: wash spring onion or leek in water

4. General cooking activity: season the steak

5. Make coffee: make coffee:

6. Stir and mix ingredients on kitchenware to cook: stir coffee or tea

7. Make bread: serve bread

8. General cooking activity: operate range hood

9. Kitchen cleaning activity: put cooking tools, dishes, or utensils in a kitchen sink

10. Peel and cut ingredients: cut pizza

11. Making multiple recipes: wash tomato in water

12. General kitchen activity: serve soup dish

13. Make pasta: remove seeds from ingredients

14. Cook in an oven: preheat the oven

15. Kitchen cleaning activity: wipe dishes and utensils with clothes

16. General cooking activity: toast bread

17. Make cookie: bake for 10-12 minutes or until golden brown

18. Make taco: remove seeds from ingredients

19. Kitchen cleaning activity: wash dishes and utensils in dishwasher

20. Make recipes: flip the flatbread to cook the other side

21. Peel and cut ingredients: cut herbs

22. General cooking activity: drain noodle

23. General activity: clean nose

24. Peel and cut ingredients: cut bacon, sausage, or ham

25. Make baked goods: score the dough for bake

26. Make recipes: make cake batter

27. Kitchen cleaning activity: wash knife in kitchen sink

28. Stir and mix ingredients on kitchenware to cook: stir zucchini

29. Make western breakfast: serve bread

30. Make pasta: wash carrot in water

31. Cook on a stovetop: remove a pan or pot on the stovetop

32. Making multiple recipes: serve egg dish

33. General cooking activity: taste ingredients or recipe

34. General kitchen activity: serve rice dish

35. Make stew or soup: remove seeds from ingredients

36. General cooking activity: check the doneness of recipe

37. Add ingredients to the recipe: add zucchini to recipe

38. General cooking activity: soak sliced onions in water

39. Add ingredients to the recipe: add potato to recipe

40. Cook on a wood stove: check and adjust the heat on a wood stove

41. General kitchen activity: serve drinks

42. Make baked goods: operate dough mixer

43. Stir and mix ingredients on kitchenware to cook: stir bean sprout

44. Make baked goods: add fillings into the dough

45. Make flatbread: coat the dough with flour

46. Kitchen cleaning activity: add dish soap to a dish to wash

47. General cooking activity: cook with or prepare milk

48. General kitchen activity: take cooked ingredients out from a wok or large pan

49. Add ingredients to the recipe: add onion to recipe

50. Add ingredients to the recipe: add mushroom to recipe

51. General kitchen activity: open the lid of a pot or pan

52. Add ingredients to the recipe: add rice to a pot

53. Make noodle soup: cut meat or seafood

54. Cook on a stovetop: adjust a pan or pot on the stovetop

55. Peel and cut ingredients: cut banana

56. General cooking activity: flip ingredients on a pan to cook the other side

57. Make pasta: wash dishes and utensils in kitchen sink

58. Add ingredients to the recipe: add meat to recipe

59. Cooking common: crack fried dough

60. General activity: harvest crops, fruits, or vegetables

61. Making multiple recipes: remove ingredients into the oven

62. Make recipes: make tea

63. General cooking activity: wash bell pepper in water

64. Peel and cut ingredients: cut cucumber

65. General kitchen activity: cover kitchen items or ingredients with a lid

66. Add ingredients to the recipe: add water to a cup

67. Add ingredients to the recipe: add cucumber to recipe

68. Make recipes: cook rice

69. Make baked goods: check dough texture

70. Make recipes: make egg salad

71. Make toast with toppings: wash lettuce or cabbage in water

72. Making multiple recipes: stir mushroom

73. Kitchen cleaning activity: put items in drying rack

74. General activity: adjust camera

75. Add ingredients to the recipe: add pepper to recipe

76. General activity: wipe hands

77. Stir and mix ingredients on kitchenware to cook: stir sauce

78. Make flatbread: wash dishes and utensils in kitchen sink

79. Make meat marinade: mix the ingredients together with a spoon until they are fully incorporated

80. Stir and mix ingredients on kitchenware to cook: stir rice dish

81. Peel and cut ingredients: cut eggplant

82. Peel and cut ingredients: peel potato

83. Make recipes: make omelet

84. Make recipes: make sandwich

85. Peel and cut ingredients: peel the outer skin of ingredients

86. Kitchen cleaning activity: wash dishes in kitchen sink

87. Make pasta: serve noodle or pasta dish

88. Stir and mix ingredients on kitchenware to cook: stir potato

89. Make baked goods: add dry ingredients to the dough or batter

90. Make baked goods: deep fry donuts and pastries

91. General kitchen activity: dispose an item in the trash bin

92. General activity: play music

93. Make baked goods: add sauce over the dough

94. Cook on a stovetop: turn on the stovetop

95. General cooking activity: remove dirt from ingredients

96. Add ingredients to the recipe: add water to a container

97. Add ingredients to the recipe: add cabbage to recipe

98. Peel and cut ingredients: grind the spice

99. Make baked goods: add flour or baking power to recipe

100. Make baked goods: add icing paste to the dough

101. General kitchen activity: serve salad

102. General activity: non cooking miscellaneous

103. Peel and cut ingredients: cut okra

104. General cooking activity: marinate ingredients

105. Cook on a wood stove: start the fire in the wood stove

106. Make baked goods: mix ingredients in the dough mixer

107. Cook on a stovetop: check and adjust the heat on the stovetop

108. General activity: write a note

109. Make recipes: cook or prepare meat dishes

110. Make baked goods: sprinkle flour onto the cooking surface

111. Make recipes: cook or prepare bread or baked goods

112. Stir and mix ingredients on kitchenware to cook: stir soup or stew dish

113. Make bread: remove excess oil from a pan

114. Make baked goods: roll out the dough on a floured surface

115. General cooking activity: strain or drain liquids from ingredients

116. General kitchen activity: set the dining table

117. Make recipes: cook or prepare beans and lentils

118. Make recipes: pour pancake batter onto the griddle and cook

119. Make recipes: make an egg mixture

120. Make baked goods: roll out the dough using a machine

121. Add ingredients to the recipe: add spices to recipe

122. Make baked goods: shape the dough into balls

123. General cooking activity: warm up milk

124. Boil ingredients in water: boil noodles in water

125. Cook on a wood stove: remove ingredients from the microwave

126. General cooking activity: move pan for miscellaneous reasons

127. Peel and cut ingredients: cut garlic

128. Make recipes: evenly cook flatbread on both sides in a skillet over medium heat

129. Cook in an oven: arrange ingredients in oven tray or baking pan or baking sheet

130. Harvest vegetables: place the vegetables in a container for storage

131. Add ingredients to the recipe: add water to a pot

132. General cooking activity: get ingredients from pantry or shelf

133. Peel and cut ingredients: cut avocado

134. Add ingredients to the recipe: add sauce to recipe

135. Make baked goods: put dough on kneading table

136. Add ingredients to the recipe: add seasoning to recipe

137. Make baked goods: add wet ingredients to the dough or batter

138. Make flatbread: put dough into flour

139. Make baked goods: cut the dough into desired shapes

140. General kitchen activity: arrange baking sheet

141. Make baked goods: sprinkle flour on the dough

142. Make baked goods: transfer cooked flatbread to a plate or a bowl

143. General cooking activity: coat ingredients in a dry mixture

144. General activity: set the timer

145. Make recipes: cook puffed rice over high heat

146. Stir and mix ingredients on kitchenware to cook: stir noodle dish

147. Add ingredients to the recipe: add soy sauce to recipe

148. Make baked goods: fill a piping bag with the batter

