# OpenReview forum: "LLMs as a Knowledge Source for Procedural Video Understanding"
_TMLR — Rejected by TMLR_

### Review · Reviewer_Fypk · 2024-09-19

**Summary Of Contributions:**

The paper proposes a rather simple yet seemingly effective idea: use an LLM to extract relevant detailed (procedural) information about given tasks, augment human annotations with the ones provided by the LLM and use that to further improve the procedural video recognition performance of such models. The paper proposes a rather structured approach for extracting most relevant information (such as reasons, conditions, etc.) from the LLMs to maximize the impact. The training strategy is straightforward and the impact of the proposed approach is analyzed across a suite of experimentation.

**Audience:**

Yes

**Claims And Evidence:**

Yes

**Requested Changes:**

- Please elaborate on what K in Recall@K means (Page 6).
- The order of appearance of the table does not match the chronological order of explanations - this creates confusion for the reader. On a related note, I still fail to find out where Figure 8 is addressed in the text (I just found out in Section 3.4 ... this order be confusing).
- The paper would benefit from another proof-read round. Minor one: what do you mean by "procedureness"? I would break it down into multiple words but more telling.
- Why would you drop the original points in Fig. 6 and plot only linear regressions? The linear regression can be a bit deceptive in this context. Why not summarizing the improvements in a table and providing std's across multiple iterations of the same experiment?

**Strengths And Weaknesses:**

**Strengths**

- The paper is well written with a coherent narrative.
- Experimental setup and the associated tasks underneath are well designed and informative.
- The idea is simple and the results seem to be favorable.

**Weaknesses**

- The impact of the proposed approach seems to be marginal or none when it comes to the Goal (omnivore features) in Table 1. Why are more recent baselines performing worse here, and Captioning in the zero-shot setting? Please further elaborate on this.

- Further, the explanations associated with Table 1 (Page 9, Goal-Step) are confusing; where can I find the substantial improvements? which gains are the 50% and 15% referring to? Please rephrase this paragraph as a whole and further clarify the key points and takeaways.

- In Table 4, on the second col. (Step), the mAP seems to be almost invariant to the percentage of groundable text, except for all data included (proposed approach), . This is rather different for Goal, where the dynamic range is much larger. Is this due to the procedural nature of the Goal task?

- What happens if we define another naive baseline, in between the "baseline" (no extra annotation) and the LLM (structurally crafted), where e.g. you query some random text related to the task from Google (not answering how to's, but just related such as synonyms) and take the first N words and augment those as extra annotation. What if you add N synonyms to the task or N antonyms ... What I miss to capture from the experiments is to what extent relevance to the task matters? Table 4 says if you add text with negative similarity scores, you get as much as of a gain as that of positive similarity scores. This is counter-intuitive, no?

- On a related note to my previous comment, does the quality of the LLM itself matter ... what if you use BERT? This could be another ablation on the backbone LLM and it's capacity ... and where one fails.

---

> ### Author Response · Authors · 2024-11-18
> **Rebuttal to Reviewer Fypk**
>
> We would like to thank Reviewer Fypk for taking their time to review the paper. We are glad they find our paper "well written" and "well designed and informative" and that they found the ablation studies to be "thorough". Next, we address their questions and concerns:
>
> - **Goal results**. We believe zero-shot experiments in the Goal setting are significantly noisy. The goal setting, because it encompasses much larger temporal spans than the step setting, has significantly fewer annotations in the Goal-Step dataset. In the zero-shot case, this is exacerbated, and in total there are only a handful of samples that are in the zero-shot Goal test set. We believe the zero-shot Step results (which combined step and sub-step samples) are more reliable in terms of noise.
>
> - **Clarification of Table 1 on Page 9**. We updated the paper to clarify the explanations
>
> - **Variance of Step vs Goal in Table 4**. Overall, the Goal evaluations have more variance than Step evaluations, because they contain a smaller number of samples (both in train and in test). We believe this may be a cause of the higher variance in results when comparing to the Step case. Additionally, as the reviewer suggested, we believe there may be a better benefit of using our approach when the action is more specific and it contains all levels of hierarchical activity. This makes it easier to predict what is going on at the specific time and therefore the synchronization between text and video is cleaner. This may play a role in the consistency of the Step evaluation.
>
> - **Random text**. We address this specific question in the paper, in Section 4.5, based on the exact observation the reviewer is making, as we wanted to study the problem further. There, we ask the LLM to generate a poem loosely based on the annotations. Figure 7 shows that this results in much worse results, even compared to the baseline, because this approach just adds noise to the training, and no additional information.
>
>     This shows that the text with negative scores is *not* equivalent to random text, but it adds useful information that the model uses to learn about the activity. For example, the model could learn to associate "flour" with the action of making bread, even when flour is not in the specific training scene, and then use that knowledge to predict "making bread" when it sees flour, because it has learned that correspondence during training. This observation, which may seem counter intuitive, would probably be harmful if the task was different, such as object detection, where hallucination of "flour" would be a problem. However, in the specific case of hierarchical action understanding, it proves to be a benefit.
>
> - **Quality of the LLM**. Initial experiments were on the Llama 8B model resulted in unsatisfactory results, so we moved to the 70B model. This shows that the quality of the LLM itself matters. Not only that, but our task is focused on *knowledge* extraction, not simple language model as it is understood in the traditional sense of the term. That would probably be enough if we only cared about rephrasing, but we care about adding additional information to the annotations. Current "large" LLMs store much more knowledge than, say, BERT (which is also not a generative model).
>
> - **Requested changes**. Thank you for these suggestions. We accounted for them in the updated version of the paper.

---

> > ### Comment · Reviewer_Fypk · 2024-11-24
> > **Thanks for the response! Any changes on the draft?**
> >
> > Firstly, thank you for the response.
> >
> > - Regarding suggested changes, for instance an ablation on the backbone, and the impact of random to relevant text, do you plan to include these in the updated draft ... Is the draft updated already? My suggestion would be to update the draft based on the suggested changes marking them in blue (or a different color) for ease of tracking.

---

> > > ### Author Response · Authors · 2024-11-25
> > > **Changes on the draft**
> > >
> > > Thank you for your answer. Yes, the draft has been updated.
> > >
> > > Regarding the changes mentioned by the reviewer:
> > >
> > > - **Random to relevant text**: The experiment the reviewer mentioned (if we understood it correctly, please let us know) was already an ablation in the original paper. The ablation is called "poem", and is studied in Section 4.5 (the "poem" would be the random text).
> > >
> > > - **Ablation on the backbone**. Unfortunately, we no longer have access to the necessary resources to conduct further experiments.

---

### Review · Reviewer_zcF1 · 2024-10-10

**Summary Of Contributions:**

In this work, the authors explored how to exploit large language models (LLMs) to improve the ability of models that take into account videos for reasoning about procedural activities. They proposed a framework that optimizes the training process by distilling the knowledge of LLMs. For example, it extracts pre- and post-conditions of the activity, hierarchy, information about the visual scene, and the reasons for specific actions. They addressed the task of goal-step online detection on the Ego4D dataset and text-video retrieval on the YouCook2 dataset, outperforming the different adopted baselines.

**Audience:**

Yes

**Broader Impact Concerns:**

No Broader Impact Statement is required

**Claims And Evidence:**

Yes

**Requested Changes:**

In general, I think the proposed work represents a significant contribution and can be published after a revision based on the provided suggestions. I recommend revising the sections discussed in the Weaknesses and providing a clearer explanation of the issues I reported.

**Strengths And Weaknesses:**

Strengths

Understanding human behavior and intentions while performing procedural activities is a very interesting task that enables the creation of intelligent assistants to support humans.
appreciated the idea of analyzing and understanding how textual information can be used to enhance visual models' abilities in procedural understanding.
I liked the study conducted on how to prompt LLM to extract the knowledge regarding pre and post condition, hierarchy, visual environment and reason.
A thorough ablation study was conducted to understand how this information can be useful, highlighting that it is not sufficient to only have more textual annotations.
Despite the framework being technically very simple, the conducted exploration will be relevant to the community and will improve the contribution of this paper.
Experiments conducted on two tasks, considering two state-of-the-art datasets, demonstrate that the proposed framework is useful for training a visual model to address these tasks. The proposed framework outperforms all the baselines and comparative methods for both tasks, even considering a zero-shot strategy.

Weaknesses

The Abstract and Introduction sections do not mention the types of experiments conducted or the datasets used. I strongly suggest adding this information to provide a complete overview of the paper before delving into the detailed sections.

The Related Works section appears to be outdated relative to the year of submission. All the works mentioned are excellent but not recent. I recommend revising this section to include more recent works from 2023/2024. Additionally, when discussing existing datasets that involve procedural activities, the authors should reference the following datasets, which are relevant to the state of the art in procedural visual datasets:

[1] Ben-Shabat, Yizhak, Xin Yu, Fatemeh Sadat Saleh, Dylan Campbell, Cristian Rodriguez-Opazo, Hongdong Li and Stephen Gould. “The IKEA ASM Dataset: Understanding People Assembling Furniture through Actions, Objects and Pose.” 2021 IEEE Winter Conference on Applications of Computer Vision (WACV) (2020): 846-858.
[2] Ragusa, Francesco, Antonino Furnari and Giovanni Maria Farinella. “MECCANO: A Multimodal Egocentric Dataset for Humans Behavior Understanding in the Industrial-like Domain.” Comput. Vis. Image Underst. 235 (2022): 103764.

Regarding the experiments and results, the authors claim that the proposed approach outperforms all compared methods in both seen and zero-shot scenarios, but there is an evident superiority of Lavila in zero-shot. How can this be justified? I suggest the authors clarify this result further.

It is unclear why the authors used NLP tools to build a "rephraser" baseline since Lavila has a module called "Rephraser" that does the same. I expect to see Lavila's Rephraser used as a baseline for comparison with other methods.

Figure 5 is hard to understand. If classes have been combined into groups of 100 elements, why are the dots in the plot between two groups? How can we observe from the plot that the rarer the classes, the higher the benefits?

---

> ### Author Response · Authors · 2024-11-18
> **Rebuttal to Reviewer zcF1**
>
> We would like to thank Reviewer zcF1 for taking their time to review the paper. We are glad they find our paper "very interesting" and that they found the ablation studies to be "thorough". Next, we address their questions and concerns:
>
> - **Mention of experiments in abstract and introduction**. We added the mention of the tasks in both of these sections in the reviewed document. Thanks for the suggestion.
>
> - **Additional related work**. We updated the related work section to incorporate the suggested references, as well as additional ones. See the updated paper.
>
> - **Zero-shot results**. We believe zero-shot experiments in the Goal setting are significantly noisy. The goal setting, because it encompasses much larger temporal spans than the step setting, has significantly fewer annotations in the Goal-Step dataset. In the zero-shot case, this is exacerbated, and in total there are only a handful of samples that are in the zero-shot Goal test set. We believe the zero-shot Step results (which combined step and sub-step samples) are more reliable in terms of noise.
>
> - **Rephraser**. For fairness in the comparisons, we decided to use the same language model for rephrasing as we used for the other tasks. Lavila's rephraser model is an older T5 model. Other than that, we use the model in the same way it is used in Lavila, therefore replicating their method in our setting.
>
> - **Figure 5**. The dots represent the range. For example, classes that have between 0 and 100 occurrences are represented with the first (leftmost) dot. Classes that have between 101 and 200 occurrences are represented with the second dot, and so on. The rarer classes (this is, the classes with fewer occurrences in the training data) are represented at the left side of the x-axis. We can see that those benefit the most from our approach (the relative improvement in accuracy is larger), compared with the most common classes (higher number of occurrences in the training data), which are represented at the right part of the x-axis in the figure. We updated the caption in Figure 5 to reflect these clarifications.

---

### Review · Reviewer_8den · 2024-10-19

**Summary Of Contributions:**

The paper proposes to leverage the knowledge of LLMs as a form of weak supervision for procedural video learning. To that extend, the authors propose to prompt LLMs for a detailed description of procedural tasks. Then the respective generated text is matched with pre-extracted video features via InfoNCE, but no further detailed information is given beyond that, e.g. which temporal range is matched etc. The setup is evaluated on two datasets, Goal-Step and YouCook2.

**Audience:**

No

**Broader Impact Concerns:**

There are not concerns about ethical implications.

**Claims And Evidence:**

No

**Requested Changes:**

I'm not sure if I can provide a comprehensive list of changes here. It would be the work of the authors to figure out how to shape their research in a way that it provides insights for the community.

**Strengths And Weaknesses:**

Strenghts:
-The idea of augmenting exiting information resp. weak annotations with LLMs is very timely and valuable for the video community.

Weaknesses:
- I could no identify any substantial novelty in the proposed approach. The idea of augmenting annotations with LLMs is already very widespread and there is no comparison how the method distinguishes for other approaches in the field. The loss function is standard, and further details are not provided.

- The evaluation is very limited. The performance on the step goal dataset seems reasonable, but the performance on YouCook2 is significantly below SotA. while this makes sense as the system is trained from scratch on small data, it does not allow for a fair SotA comparison and is thereful only medium helpful in drawing any conclusions about the value of the proposed method.

- Missing technical details: There are not many details given about the data and the training itself. E.g. which part of the video is match which which text embedding.

- Missing evaluation: The evaluation mainly focuses on the language model, but even here, the language model is fixed, so, so far, we only learn about how a specific language model would react under those specific conditions. It would be good to compare at least output of different language models. Also, InfoNCE seems a weird choice if abundant text is available. It would be good to also consider MIL-NCE.

---

> ### Author Response · Authors · 2024-11-18
> **Rebuttal to Reviewer 8den**
>
> We would like to thank Reviewer 8den for taking their time to review the paper. We address their concerns next:
>
> - **Novelty**. While LLMs have been used to augment annotations in the literature, they have not explicitly focused on extracting stored knowledge associated with human actions. They have mostly been used to augment data in a rephrasing way, used more as a "text" augmentation (how to say the same thing in different way), not as a knowledge information (how to add new information that was not in the original caption in the first place). In this regard, we show how adding hallucinated information (as it is not necessarily present in the image) is not only not hurting, but it contributes to our task, which is specific for the hierarchical human action studies and particular in the sense that it is not the case in most other scenarios.
>
> - **Evaluation**. We believe the Goal-Step dataset is the most adequate to evaluate the benefits of our approach, as it is specifically designed to exploit the hierarchical nature of actions. Other datasets do not convey this hierarchical nature of actions as accurately.
>
> - **Technical details**. The existing text annotations have specified time spans (e.g. from second 20 to 45 in the video). They are considered positives for all visual samples within that span. We clarified this in Section 3.4 of the updated paper.
>
> - **InfoNCE vs MIL-NCE**
>
>     MIL-NCE assumes that the temporal alignment between the text and the video may be noisy, as the data they use is collected automatically from YouTube videos. In our scenario, we do not have the same problem because the datasets we use are manually annotated.
>
>     Additionally, in the spirit of MIL-NCE, we did try an experiment where instead of a single LLM-generated annotation, we collected N (N=5 in our experiment), and only backpropagated the loss through the sample with the largest similarity. Note that these samples do not assume noise in the time synchronization, but in the LLM generation process. We found no strong benefit of this approach, and because it was N times more expensive to run, we did not use it further.
>
>     Finally, note that if a training dataset was used where temporal alignment was a concern, our approach is perpendicular to the choice of a loss function (as the reviewer mentioned, it is not part of our contribution) and therefore it would be extremely easy to use MIL-NCE as the loss function.

---

### Decision · Action_Editor_ZmxY · 2025-01-14

**Recommendation:** Reject

**Comment:**

See in Claims And Evidence.

**Audience:**

Yes.

**Claims And Evidence:**

The paper focuses on procedural video understanding, which is the task of understanding activities that happen in steps (A then B then C then ...). The key idea of the paper is to use LLMs so as to describe the procedural actions in an expanded manner, that the video model can leverage as additional supervision to recognize activities better. As described by figure 2, there is an LLM that receives as input the ground truth label of the action and a prompt to create a caption that describe the procedural action. This extended description is passed through an LLM again, whose features are then compared with the features of the video model, whose ground truth label was used. The intuition is that the more refined LLM-based textual description will guide the video model to learn features that are better aligned with the ground truth task. Clearly, the hope is that different procedural tasks share steps, and like that the video model can amortize the learning and learn more general features, that lead to higher accuracies.

Reviewers were lukewarm positive. The main remarks were:
- lack of clear substantial contribution (8den).
- modest results on Goal (Fypk) and unclear improvements in the zero-shot case (Youcook2). Overall, a comment that evaluation seems incomplete (8den), considering that only one type of language models was used only.
- Requirement for strong alignment between video, and text

I tend to agree that the idea is simple (and by itself that is not bad), however, the validation is modest, to say the least. Only 2 datasets are compared, and the main results for one of them (table 1, step) casts shadows onto the model. The authors claim that the specific setting in the dataset is not suitable, but in that case the paper validates only in 1.5 datasets.

I also note that most comparisons are with methods that are (by today's standards) old-ish, although there is significant improvement compared to Song et al. from 2023.

Overall, given that this is clearly an experimental paper, I would expect much more substantial evidence to support an acceptance.

**Resubmission Of Major Revision:**

The authors may consider submitting a major revision at a later time.